# Over-activation of primate subgenual cingulate cortex enhances the cardiovascular, behavioral and neural responses to threat

Laith Alexander[1,2,4,6], Christian M. Wood [1,2,6], Philip L. R. Gaskin [1,2,5], Stephen J. Sawiak[2,3], Tim D. Fryer[3], Young T. Hong[3], Lauren McIver[1,2], Hannah F. Clarke[1,2] & Angela C. Roberts [1,2]✉

Stress-related disorders such as depression and anxiety are characterized by enhanced negative emotion and physiological dysfunction. Whilst elevated activity within area 25 of the subgenual anterior cingulate cortex (sgACC/25) has been implicated in these illnesses, it is unknown whether this over-activity is causal. By combining targeted intracerebral micro-infusions with cardiovascular and behavioral monitoring in marmosets, we show that over-activation of sgACC/25 reduces vagal tone and heart rate variability, alters cortisol dynamics during stress and heightens reactivity to proximal and distal threat. [18]F-FDG PET imaging shows these changes are accompanied by altered activity within a network of brain regions including the amygdala, hypothalamus and dorsolateral prefrontal cortex. Ketamine, shown to have rapid antidepressant effects, fails to reverse elevated arousal to distal threat contrary to the beneficial effects we have previously demonstrated on over-activation induced reward blunting, illustrating the symptom-specificity of its actions.

[1] Department of Physiology, Development and Neuroscience, University of Cambridge, Cambridge CB2 3DY, UK. [2] Behavioural and Clinical Neuroscience Institute, University of Cambridge, Cambridge CB2 3EB, UK. [3] Wolfson Brain Imaging Centre, Department of Clinical Neurosciences, University of Cambridge, Cambridge CB2 0QQ, UK. [4] Present address: St. Thomas' Hospital, London, UK. [5] Present address: Benevolent AI, London, UK. [6] These authors contributed equally: Laith Alexander, Christian M. Wood. ✉email: acr4@cam.ac.uk

I dentifying the neurobiological basis of enhanced negative emotion in stress-related disorders such as depression and anxiety is a major challenge in neuroscience and is crucial in guiding future treatment strategies. Countless correlative human neuroimaging studies have related dysfunctional activity within the ventromedial prefrontal cortex (vmPFC) to impairments in cardiovascular, endocrine and behavioral indices of negative emotion, but there are two important outstanding issues. First, it is unknown whether these activity changes are causal, and second, if causal, whether this region acts to promote or inhibit stress-related symptoms.

It is critical to recognize that the vmPFC is a large and likely functionally heterogenous region, variably including caudal subgenual anterior cingulate cortex (sgACC, area 25); rostral sgACC and medial PFC (areas 14, 10, 24); and pregenual regions of the anterior cingulate (pgACC, area 32)[1–5]. Consistent with functional heterogeneity, activity within these subzones has been linked to several opposing manifestations of negative emotion and its regulation. For example, sgACC comprises a large portion of the vmPFC and is associated with threat generalization[6,7] but also threat extinction[8,9]. An overlapping region (including areas 10, 14, and 25) signals stress controllability[10,11] and regulates emotion-generating structures following stress exposure[12]. Area 10 specifically has been linked to intolerance of uncertainty during fear extinction[13].

A similar portion of the vmPFC is also implicated in the central regulation of autonomic function, forming part of a central autonomic network which functions to maintain a constant internal milieu[14–16]. Recently, our laboratory implicated sgACC/25 but not pgACC/32 in the regulation of basal cardiovascular activity: temporary inactivation of sgACC/25 in marmosets increased heart rate variability (HRV) by elevating vagal (parasympathetic) tone[17]. The question remains as to whether the over-activity in sgACC/25 reported in neuroimaging studies of people with depression and anxiety[5,18–20] is causally related to their cardiovascular and emotional dysfunction, and if so, how?

Studies in non-human primates (NHPs) are essential in addressing questions of causality since, compared to rodents, the structural organization of the vmPFC in NHPs is far closer to that of humans[21,22]. The differences between rodent and primate are highlighted by our recent finding that inactivation of marmoset sgACC/25 and pgACC/32 has opposing functional effects to that seen following inactivation of their putative rodent anatomical homologs (infralimbic cortex and prelimbic cortex respectively)[17]. Indeed, the wide and persistent translational gap between advances in preclinical research and a relative failure to develop more effective treatments for psychiatric disorders is due, in part, to a lack of understanding of the complex control of negative affect exerted by the highly evolved primate vmPFC.

Previously, in the domain of reward, we have shown that over-activity of marmoset sgACC/25 induces anticipatory and motivational aspects of anhedonia-like behavior, and that ketamine—shown to have rapid antidepressant effects[23]—can ameliorate these deficits[24]. However, what of the effects of sgACC/25 over-activity in the negative domain? To address this critical question the present study takes a comprehensive, multi-dimensional approach in marmosets to determine whether causal links exist between over-activity in sgACC/25, physiological dysfunction and enhanced negative reactivity to threat (Fig. 1).

Using targeted intracerebral microinfusions of the excitatory amino acid transporter-2 (EAAT-2) inhibitor dihydrokainic acid (DHK) to reduce glutamate re-uptake from synapses, we first ascertained the impact of sgACC/25 over-activation on cardiovascular and endocrine (cortisol) dynamics in an emotionally neutral condition. We then assessed the effect of sgACC/25 over-activation on several indices of affect regulation during both proximal and distal threat as defined by the predatory imminence theory, in which threat engages distinct cognitive and behavioral strategies depending on its closeness in time and space[25,26].

The first test of the regulation of proximal threat involved a Pavlovian conditioned threat and extinction paradigm with a rubber snake as an ethologically relevant unconditioned stimulus (US). This paradigm was used to investigate the effects of sgACC/25 over-activation on the extinction of a conditioned threat response, and the subsequent recall of that extinction memory the next day. The second test focused on the ability of animals to discriminate between threatening and safety cues during an aversive Pavlovian discriminative conditioning paradigm. We utilized positron emission tomography (PET) imaging of the glucose analog 2-deoxy-2-[18F]fluoro-D-glucose (18F-FDG) during this paradigm to determine the circuit-wide changes associated with sgACC/25 over-activation. Finally, we looked to replicate our previous finding of sgACC/25 over-activation enhancing arousal to distal threat in the form of a human intruder[24]. The novel question we addressed here was whether this effect could be ameliorated by ketamine, in the same way we had previously demonstrated that ketamine could ameliorate the blunting of appetitive anticipatory arousal.

## Results
An overview of the marmosets used in these experiments is shown in Table 1 with an experimental timeline in Fig. 1a and illustrations of three of the main paradigms in Fig. 1b–d. Histological assessment of cannula placement is shown in Fig. 1e, together with estimated spread of infusions[27]. All sgACC/25 infusions were bilateral. In all conditions, cardiovascular measurements were made using wireless telemetric monitoring with an aortic catheter probe.

**sgACC/25 over-activation impacts basal cardiovascular function by increasing heart rate and reducing heart rate variability, without affecting blood pressure or cortisol levels.** To determine the effects of sgACC/25 over-activation on basal cardiovascular function, seven marmosets were assessed in a familiar, affectively neutral environment in the absence of any cues (Fig. 1b). Over-activation of sgACC/25 with DHK had no effect on blood pressure (mean arterial pressure [MAP]; Fig. 2a) but significantly increased heart rate (Fig. 2b, c), reduced HRV (Fig. 2d) and altered the sympathetic-parasympathetic balance as measured by an increase in the ratio between the cardiac sympathetic index (CSI) and cardiac vagal index (CVI, Fig. 2e). There was a shift towards sympathetic predominance, with a significant decrease in CVI (Fig. 2f) and increase in CSI (Fig. 2g). In three animals in which salivary cortisol levels were measured, cortisol levels did not change from before to after the testing session in both control and over-activation conditions (Fig. 2h).

**sgACC/25 over-activation disrupts the extinction of conditioned proximal threat and generally heightens cardiovascular arousal.** To investigate the effects of sgACC/25 over-activation on the extinction of Pavlovian conditioning to proximal threat, four marmosets were tested on an adaptation of the classic Pavlovian conditioned threat and extinction paradigm used in rodents (Fig. 1c). The sight of a rubber snake (snakes being a predator of marmosets and ethologically aversive) replaced an aversive foot shock as the US (as in ref. [17]). This was paired with a 15 s tone as the conditioned stimulus (CS; Supplement Fig. S1). Marmosets showed a US-induced rise in blood pressure upon each presentation of the snake (Supplement Fig. S2A), together with elevated salivary cortisol levels after snake exposure (Supplement

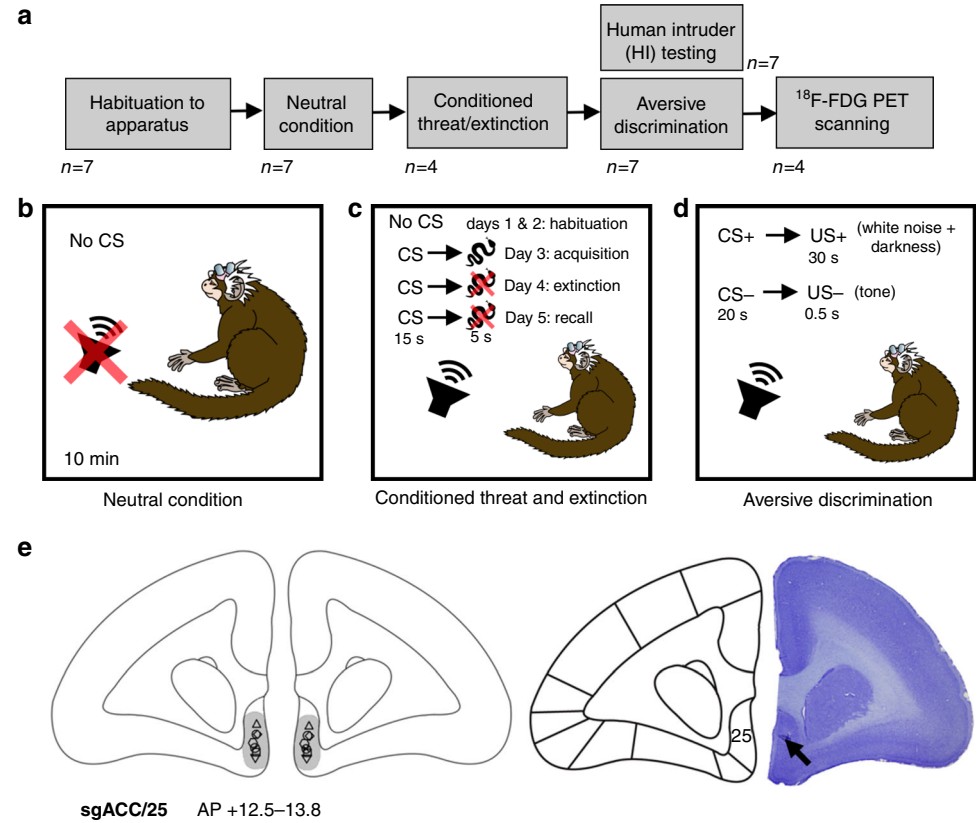

**Fig. 1 Experimental timeline, task overview, and histological assessment of cannula placements. a** Experimental timeline. All marmosets ($n = 7$) were first habituated to the testing apparatus so the context became affectively neutral, after which sgACC/25 infusions were carried out in this neutral condition to assess basal cardiovascular changes. Next, marmosets underwent infusions on the conditioned threat and extinction ($n = 4$) and aversive Pavlovian discriminative conditioning ($n = 7$) paradigms (see Table 1 for details). Human intruder (HI) testing ($n = 7$) was interleaved with aversive discrimination testing. Finally, four marmosets underwent $^{18}$F-FDG PET scanning. **b** In the neutral condition, no CSs were presented. **c** In the Pavlovian conditioned threat and extinction paradigm, habituation occured over the first two days; learning of the CS (auditory cue)-US (rubber snake) association occurred over the third acquisition day; and on the subsequent extinction and extinction-recall days (days four and five), the cue was presented without the snake. **d** In the aversive Pavlovian discriminative conditioning paradigm, marmosets learned to distinguish between a threatening auditory cue (CS+) predicting aversive white noise and darkness (US+), and a safety auditory cue (CS−) predicting a neutral tone (US−). **e** Histological assessment of cannula placement using cresyl-violet staining. No damage was noted in any animals aside from the small area of gliosis used to pinpoint cannula placement. A schematic diagram, left, shows the cannula placement for all marmosets reported in the manuscript (anteroposterior [AP] extent between +12.5 and +13.8 mm from interaural line, diagram centered around +13.0mm) together with the estimated spread of infusion in gray (0.5–1.0mm). A representative cresyl-violet stained histological section is shown right, with the cannulation site indicated by an arrow.

**Table 1 Subject involvement in each phase of the study.**

| Subject | Sex | Symbol | Number of infusions | Basal cardiovascular analysis | Conditioned threat and extinction | Aversive Pavlovian discriminative conditioning | | Human intruder testing |
|---|---|---|---|---|---|---|---|---|
| | | | | | | Infusion testing | $^{18}$F-FDG PET scanning | |
| 1 | F | ▽ | 21 | ✓a | ✓b | ✓c* | | ✓c* |
| 2 | M | △ | 19 | ✓a | ✓b | ✓c* | | ✓c* |
| 3 | M | □ | 25 | ✓a | ✓b | ✓c* | ✓d | ✓c* |
| 4 | M | ○ | 24 | ✓a | ✓b | ✓c* | ✓d | ✓c* |
| 5 | F | ▯ | 22 | ✓a | | ✓b* | ✓c | ✓b* |
| 6 | M | ◇ | 16 | ✓a | | ✓b* | | ✓b* |
| 7 | F | ⊗ | 19 | ✓a | | ✓b* | ✓c | ✓b* |

A tick indicates that the subject took part in that phase of the study, with letters a–d denoting the order of testing.
A star (*) indicates parallel testing phases. See Methods for full explanation of why not all animals were tested in all conditions.
*F* female, *M* male.

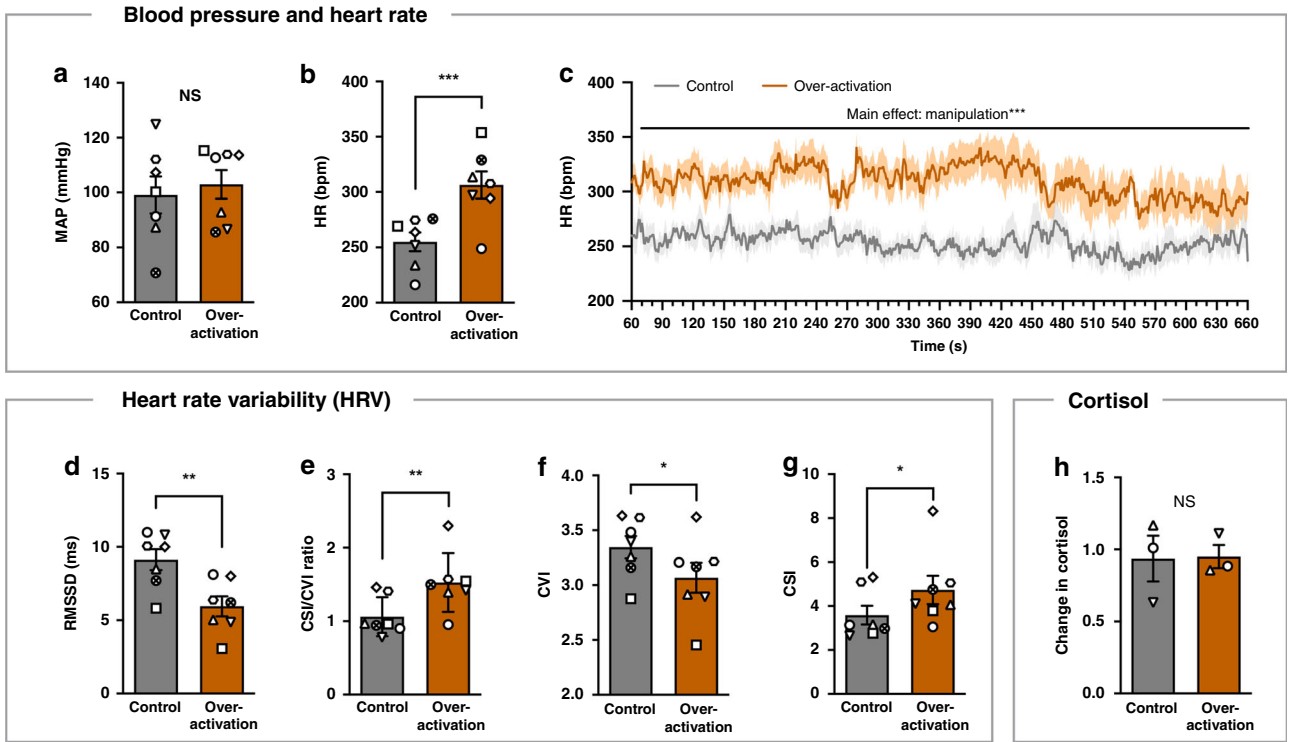

**Fig. 2 sgACC/25 over-activation shifts the sympathetic-to-parasympathetic balance, increasing heart rate and reducing heart rate variability.** Gray = control, orange = over-activation. Shading and error bars represent SEM. $n = 7$ except cortisol where $n = 3$. **a** sgACC/25 over-activation had no effect on blood pressure (mean arterial pressure, MAP) measured across the session (two-tailed paired $t$-test, $p = 0.626$). **b** sgACC/25 over-activation significantly increased heart rate (HR) measured across the session (two-tailed paired $t$-test, $p < 0.001$, $d = 2.27$). **c** Analysis of second-by-second heart rate values highlighted a systematic increase across the entire session (linear mixed-model, manipulation × time, $F < 1$, $p = 0.999$; main effect of manipulation, $F_{1,5899} = 3448$, $p < 0.001$, $d = 2.02$). **d** sgACC/25 over-activation reduced heart rate variability (HRV) as measured by the root mean square differences of successive R-R intervals (RMSSD; two-tailed paired $t$-test, $p = 0.001$, $d = 2.20$). **e** Fractionating HRV into cardiac sympathetic (CSI) and cardiac vagal (CVI) indices revealed sgACC/25 over-activation shifted sympathetic/parasympathetic balance towards sympathetic predominance (two-tailed paired $t$-test, $p = 0.004$, $d = 1.69$). **f** sgACC/25 over-activation significantly decreased CVI (two-tailed paired $t$-test, $p = 0.011$, $d = 1.38$) and **g** significantly increased CSI (two-tailed paired $t$-test, $p = 0.032$, $d = 1.07$). **h** Cortisol levels did not change from before to after the habituation session (mean ± SEM ratio: control $0.937 ± 0.159$, over-activation $0.951 ± 0.080$) and did not differ between control and over-activation conditions (two-tailed paired $t$-test, $p = 0.960$). Source data are provided as a Source Data file.

Fig. S2B). They also acquired a conditioned vigilant scanning (VS) response to the CS during the acquisition session; this effect remained stable across infusion blocks and did not differ between "to be saline" and "to be over-activation" blocks (Fig. 3a, acquisition). As expected, this response declined across the extinction session when the rubber snake was no longer presented (Fig. 3a, extinction, gray line) and showed partial re-instatement and re-extinction on the extinction recall session (Fig. 3a, extinction recall, gray line) under the control (saline) condition. A similar pattern of acquisition, extinction and extinction recall was seen with respect to conditioned blood pressure (Fig. 3b, gray lines) except that the conditioned response was not CS specific (Supplement Fig. S2C). Instead, the rise in blood pressure generalized to the context as indicated by a sustained elevation in blood pressure following the first US+ presentation (as seen by the step change in blood pressure in Supplement Fig. S2A).

Over-activation of sgACC/25 prior to the extinction session elevated both blood pressure and VS responses during extinction (Fig. 3a, b, extinction, orange lines). However, while the extinction of VS behavior was blocked following over-activation, the rate of blood pressure extinction remained the same. Salivary cortisol samples were acquired pre- and post-extinction: the ratio of post-extinction to pre-extinction cortisol was significantly higher following over-activation compared to saline infusions (Supplement Fig. S3). Therefore, while sgACC/25 over-activation

does not affect cortisol levels in basal conditions, it appears to potentiate HPA axis activity in threatening contexts. On the subsequent extinction recall day, during which no infusion took place, VS behavior remained elevated but extinguished across repeated CS presentations compared to control conditions (Fig. 3a, extinction recall, orange line). Blood pressure remained systematically higher than in control conditions (Fig. 3b, extinction recall, orange line).

The effect of sgACC/25 over-activation to elevate cardiovascular and behavioral arousal was not confined to CS periods. When blood pressure values were plotted across the entire extinction and extinction recall sessions, it was clear there was a sustained elevation in blood pressure (Fig. 3c, e). This blood pressure effect was not observed in the affectively neutral condition, highlighting the dependency on an affectively negative/threatening context. There was also a tendency for increased VS during the baseline periods (Fig. 3d, f), again indicating a generalization effect, although these changes were not statistically significant.

**sgACC/25 over-activation increases arousal when discriminating between threat and safety cues during aversive Pavlovian discriminative conditioning, and induces metabolic changes in the amygdala, hypothalamus, and prefrontal cortex.** Having shown generalization effects on the conditioned threat

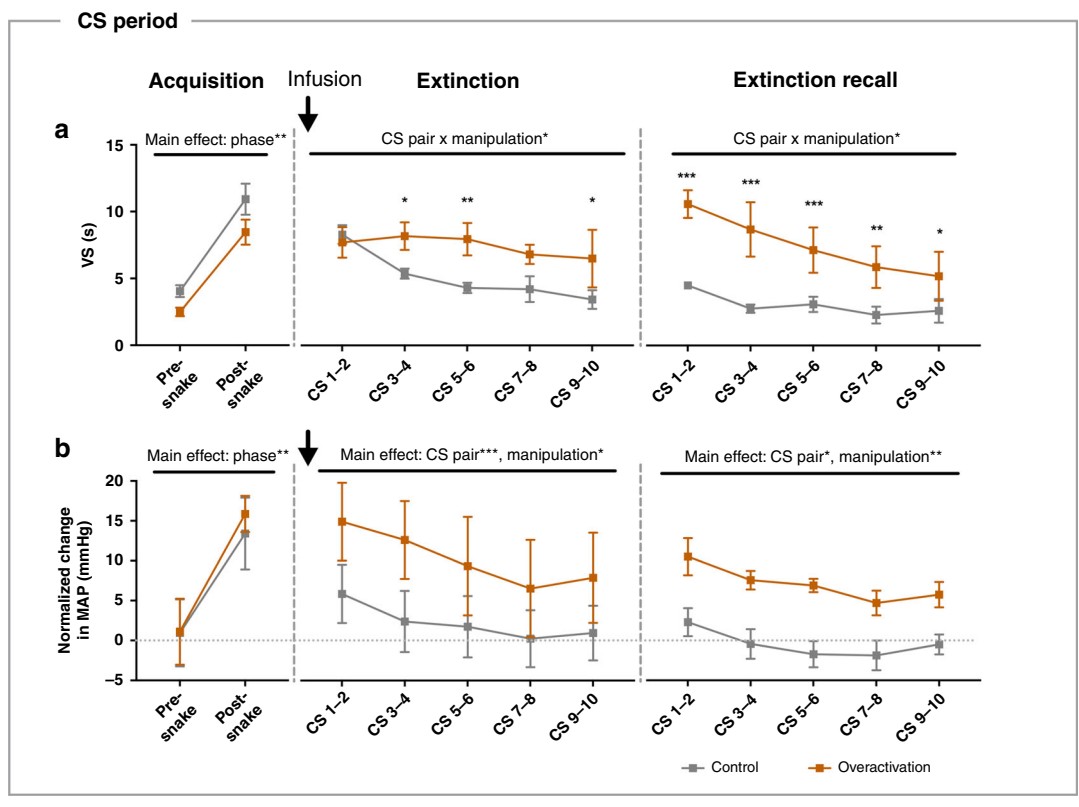

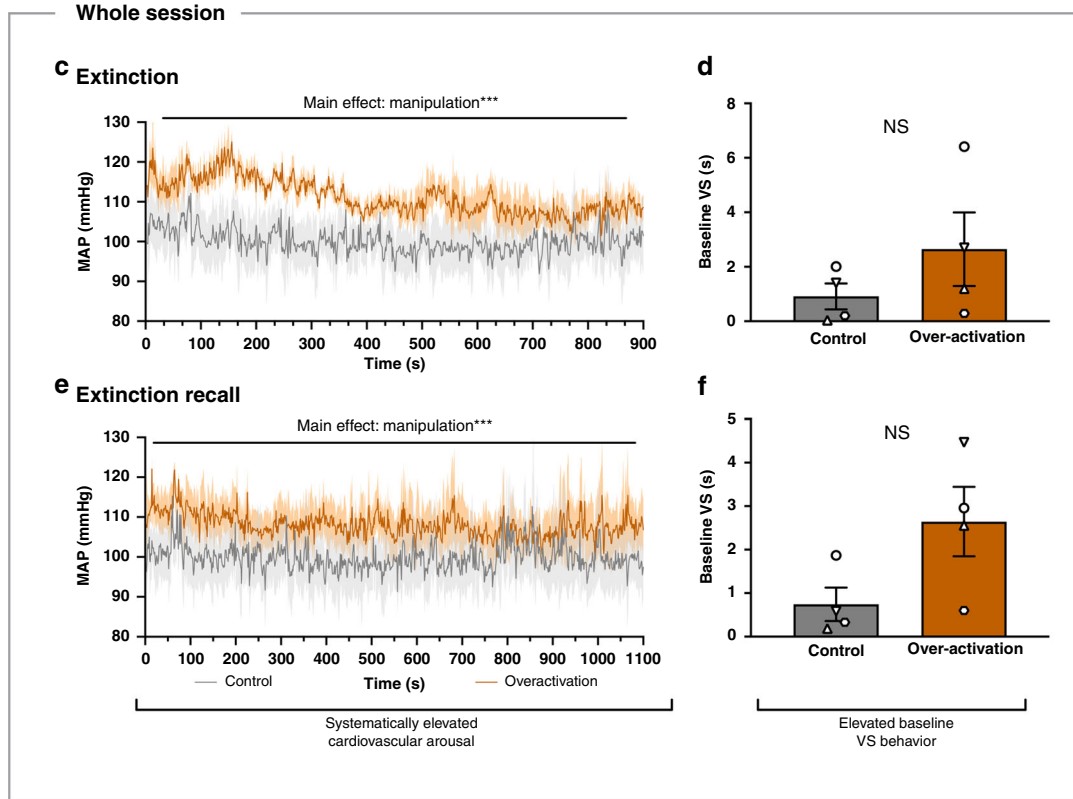

and extinction paradigm, we next investigated the effects of over-activation specifically on marmosets' ability to discriminate between threatening and safe cues. This allowed us to determine the extent and specificity of the generalization effect to proximal threats. In the aversive Pavlovian discriminative conditioning paradigm (Fig. 1d), seven marmosets were presented with two cues: a threatening auditory cue (CS+, 20 s) that predicted a period of mildly aversive darkness and unpredictable white noise (US+, 30 s), and a safety cue (CS−, 20 s), which predicted a non-aversive tone (US−, 0.5 s; Supplement Fig. S4). On infusion days, two safety cues were presented with one threatening cue in the order $CS_{-1}/CS+/CS_{-2}$.

**Fig. 3 sgACC/25 over-activation elevates threat-related arousal during threat extinction and extinction-recall.** Gray = control, orange = over-activation. Shading and error bars represent SEM. Infusions took place prior to extinction sessions, indicated by arrows above figures (**a**, **b**). $n = 4$. **a** During acquisition sessions, animals increased their vigilant scanning (VS) response from the pre-snake to post-snake phase in both the "to be saline" and "to-be overactivation" blocks (two-way repeated measures ANOVA, manipulation × phase, $F_{1,3} = 1.4$, $p = 0.317$; main effect of phase, $F_{1,3} = 36.5$, $p = 0.009$, $\eta^2 = 0.777$). sgACC/25 over-activation blocked behavioral extinction (two-way repeated measures ANOVA, manipulation × CS pair, $F_{4,12} = 3.6$, $p = 0.039$, $\eta^2 = 0.087$) with significant differences in scanning responses evident during the second (Sidak's test, $p = 0.039$, $d = 1.86$), third ($p = 0.007$, $d = 2.04$) and fifth ($p = 0.023$, $d = 1.05$) CS pairs, but not the first ($p = 0.974$) or fourth ($p = 0.0602$). These effects persisted to the next day with sgACC/25 over-activation elevating VS across extinction recall (two-way repeated measures ANOVA, manipulation × CS pair, $F_{4,12} = 4.0$, $p = 0.027$, $\eta^2 = 0.039$) evident across all CS pairs (Sidak's test, first $p < 0.001$, $d = 2.55$; second $p < 0.001$, $d = 1.39$; third $p < 0.001$, $d = 1.22$; fourth $p = 0.003$, $d = 1.08$; fifth $p = 0.026$, $d = 0.83$). **b** Equally, the blood pressure (MAP) increase during acquisition was not different in the "to be saline" vs. "to be overactivation" blocks (two-way repeated measures ANOVA, manipulation × phase, $F < 1$, $p = 0.695$; main effect of phase, $F_{1,3} = 44.5$, $p = 0.007$, $\eta^2 = 0.503$). sgACC/25 over-activation did not alter the rate of blood pressure extinction (two-way repeated measures ANOVA, manipulation × CS pair, $F < 1$, $p = 0.668$; main effect of CS pair, $F_{4,12} = 16.0$, $p < 0.001$, $\eta^2 = 0.069$) but did elevate blood pressure responses across all CS pairs (two-way repeated measures ANOVA, main effect of manipulation, $F_{1,3} = 11.8$, $p = 0.042$, $\eta^2 = 0.180$). Marmosets continued to extinguish their blood pressure responding during extinction recall (two-way repeated measures ANOVA, main effect of CS pair, $F_{4,12} = 4.2$, $p = 0.023$, $\eta^2 = 0.113$) and again there was a systematic elevation in blood pressure responses following over-activation (two-way repeated measures ANOVA, main effect of infusion, $F_{1,3} = 39.1$, $p = 0.008$, $\eta^2 = 0.562$). **c** Plotting second-by-second blood pressure values for all four animals during extinction revealed an increase across the entire session, not restricted to CS periods (linear mixed-model, manipulation × time, $F < 1$, $p = 0.999$; main effect of manipulation, $F_{1,5330} = 1897$, $p < 0.001$, $d = 1.17$). **d** Three out of four animals showed increased scanning during the baseline periods of the extinction session following over-activation, suggestive of a behavioral generalization response, but the increase was not significant (two-tailed paired $t$-test, $p = 0.182$). **e** A systematic elevation in blood pressure values was also evident for all four animals across the extinction recall session (linear mixed-model, manipulation × time, $F < 1$, $p = 0.999$; main effect of manipulation, $F_{1,6547} = 720$, $p < 0.001$, $d = 1.34$). **f** Again, three out of four animals also showed elevated baseline scanning following over-activation in the extinction recall session, but this change was not significant (two-tailed paired $t$-test, $p = 0.087$). Source data are provided as a Source Data file.

Marmosets successfully acquired CS directed conditioned VS, blood pressure and heart rate responses together with a robust blood pressure response to the US+ (Supplement Fig. S5). While sgACC/25 over-activation did not significantly impact upon animals' behavioral or cardiovascular discrimination between the cues (Fig. 4a, b), it did induce a generalized increase in arousal indicated by an elevation in VS and blood pressure responses during the baseline (BL) periods, 20 s before CS onset (Fig. 4c, d). To determine whether the recovery from a stressor (US+) was also affected by sgACC/25 over-activation, the 10 s period following US termination was analyzed. During this period, blood pressure values remained higher for longer after over-activation (Fig. 4e). This suggests that recovery from a stressor is slowed by sgACC/25 over-activation, although this finding is confounded by the baseline effects of over-activation.

To assess circuit-wide changes induced by sgACC/25 over-activation during the task, four of the seven marmosets underwent [18]F-FDG PET imaging. Each marmoset had two counter-balanced scans: one following saline infusion and one following over-activation using DHK. In all scans, marmosets were injected with [18]F-FDG and received a lengthened version of the aversive Pavlovian discriminative conditioning session in the test apparatus (Fig. 5a). They were then anesthetized and scanned. For the voxel-wise analysis, paired $t$-tests were performed in a general linear model with contrasts for increases and reductions in [18]F-FDG uptake (standardized uptake value ration (SUVR) compared to the cerebellum) following over-activation with DHK versus saline controls. We had a priori interest in regions with which sgACC/25 has strong reciprocal connectivity, including orbitofrontal cortex (OFC), frontopolar cortex, lateral PFC, medial temporal lobe, and diencephalon[28].

In the first demonstration of the brain effects of sgACC/25 over-activation in negatively valenced behavioral settings, imaging showed that over-activation increased [18]F-FDG uptake in sgACC/25 together with increases in the amygdala, hypothalamus and temporal lobe (Fig. 5b). Lower metabolic activity was found in dorsolateral prefrontal cortex (dlPFC)/46, central orbitofrontal cortex (OFC)/13, frontopolar cortex/9 and caudate (Fig. 5c). These imaging changes were accompanied by elevated behavioral and cardiovascular arousal during the task (Supplement Fig. S6).

**Over-activation induced increases in anxiety-like responses to a human intruder are not ameliorated by ketamine.** To investigate distal threat processing, we assessed the responsivity of seven marmosets to a human intruder (HI)—a distal threat in the form of an unfamiliar human, to which marmosets display a range of behaviors (Fig. 6a). As described in ref. [29], we used exploratory factor analysis (EFA) to load these behaviors onto a single latent factor representing anxiety-like responses (Fig. 6b). Over-activation of sgACC/25 enhanced responsivity of marmosets to the HI, as evidenced by an increased factor score (Fig. 6c), replicating a finding previously reported by our laboratory[24].

The heightened anxiety-like behavior to the HI following sgACC/25 over-activation provided us with an opportunity to compare the efficacy of ketamine in ameliorating this phenotype with its efficacy in ameliorating an anticipatory anhedonia-like phenotype induced by the same manipulation reported previously[24]. Marmosets were injected with 0.5 mg/kg ketamine i.m. and were tested 24 h later on the HI paradigm following sgACC/25 over-activation (Fig. 6a). This timepoint corresponds to one of the timepoints at which ketamine successfully reversed anticipatory reward blunting[24]. However, in the case of the HI test, ketamine had no effect on anxiety-like behavior—the anxiogenic effect induced by sgACC/25 over-activation was still observed (Fig. 6c). Detailed responses of animals across all conditions reported here are shown in Supplement Table S1.

## Discussion

Until now it was unknown whether the altered vmPFC activity demonstrated in correlative neuroimaging studies of mood and anxiety disorders could cause dysregulation of negative emotion, and if so, where the critical locus was located within vmPFC and the direction of its effects. The findings here directly address these questions by showing that over-activation of the caudal subgenual region of primate vmPFC, sgACC/25, reduces basal HRV and enhances both proximal and distal threat-evoked cardiovascular and behavioral arousal mirroring the changes observed in stress-related disorders. We have previously demonstrated the selective effects of sgACC/25 manipulations compared to adjacent brain regions[17,24], highlighting that this causal link is specific to sgACC/25.

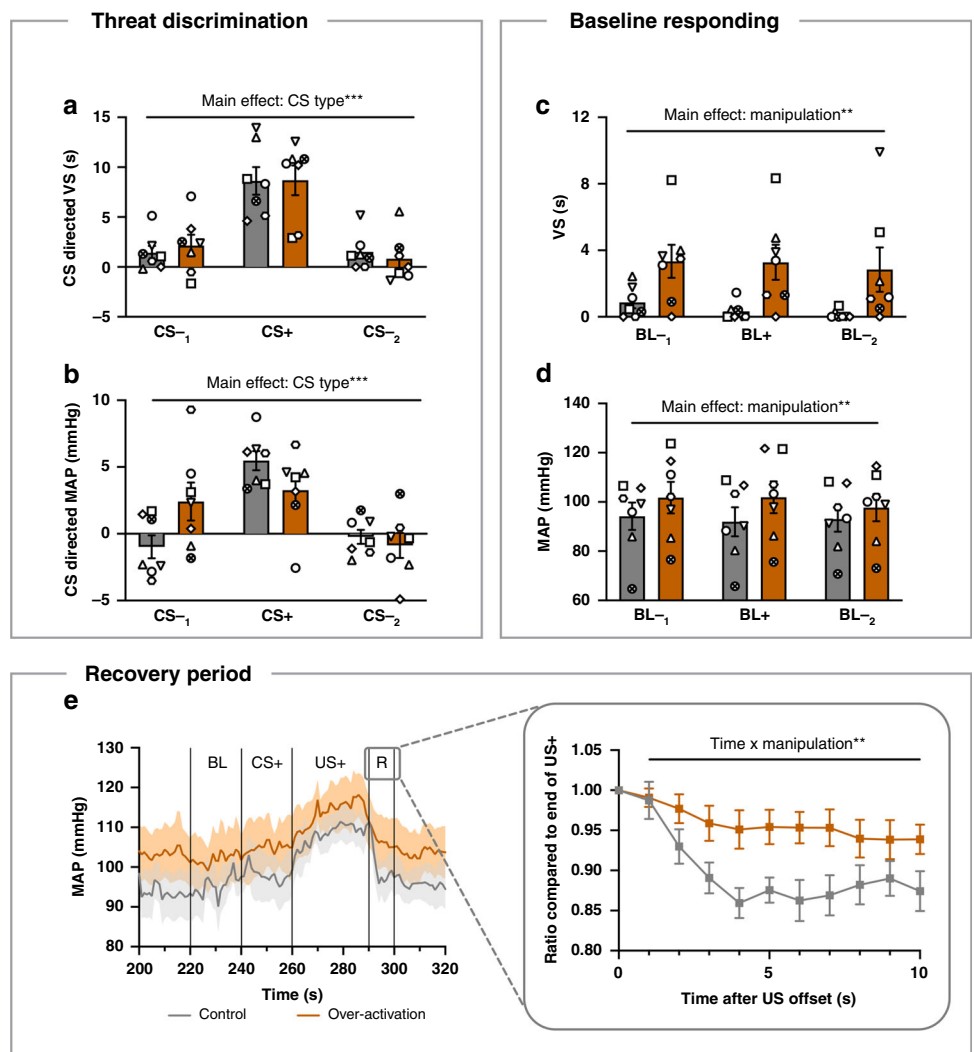

**Fig. 4 Intact discrimination between threat and safety cues, but elevated baseline responding and slowed stress recovery, following sgACC/25 over-activation.** Gray = control, orange = over-activation. Shading and error bars represent SEM. BL indicates baseline (20s pre-CS). $n = 7$. **a** sgACC/25 over-activation did not alter behavioral discrimination between safety (CS$_{-1}$ and CS$_{-2}$) cues and a threatening (CS+) cue (two-way repeated measures ANOVA, manipulation × CS, $F < 1$, $p = 0.647$; main effect of CS type, $F_{2,12} = 32.3$, $p < 0.001$, $\eta^2 = 0.619$). **b** There was a trend towards flattening of the blood pressure (MAP) discrimination between safety cues and a threatening cue following over-activation (two-way repeated measures ANOVA, manipulation × CS, $F_{2,12} = 3.8$, $p = 0.054$; main effect of CS type, $F_{2,12} = 19.5$, $p < 0.001$, $\eta^2 = 0.382$). **c** sgACC/25 over-activation elevated baseline VS (two-way repeated measures ANOVA, manipulation × CS, $F < 1$, $p = 0.909$; main effect of manipulation, $F_{1,6} = 8.4$, $p = 0.027$, $\eta^2 = 0.307$). **d** sgACC/25 over-activation also elevated baseline blood pressure (two-way repeated measures ANOVA, manipulation × CS, $F_{2,12} = 2.5$, $p = 0.124$; main effect of manipulation, $F_{1,6} = 22.7$, $p = 0.003$, $\eta^2 = 0.062$). **e** Trace showing mean blood pressure responses during the baseline, CS, US and recovery (R) phases of the CS+ trial. The recovery period was defined as the 10s period following termination of the US+. sgACC/25 over-activation slowed the decline in blood pressure (measured as a ratio of the blood pressure to the final second of the US+, inset; two-way repeated measures ANOVA, manipulation × time, $F_{9,54} = 3.0$, $p = 0.005$, $\eta^2 = 0.049$). Source data are provided as a Source Data file.

In affectively neutral conditions, sgACC/25 over-activation significantly increased basal heart rate and reduced HRV, an effect mediated by a shift in the parasympathetic-to-sympathetic balance similar to that apparent in mood and anxiety disorders. The perigenual and subgenual portions of the vmPFC have been related to vagal reactivity previously. For example, variation in the high-frequency component of HRV (thought to reflect mainly parasympathetic tone) is strongly correlated with pg/sgACC BOLD signal[30,31]. However, it is important to note that the areas of activation in such studies typically correspond to areas 10, 24, and 32—not area 25. We have shown here that there is a causal relationship between elevated sgACC/25 activity and reduced parasympathetic tone, consistent with previous findings from our group with respect to inactivation[17].

These results are also consistent with the anatomical connectivity of sgACC/25 identified from tracing studies. Neurons from primate sgACC/25 project directly to the brainstem but also indirectly via the hypothalamus and amygdala indicating that sgACC/25 has dual access to an emotional-visceral motor system[28,32–34]. Consistent with this, the $^{18}$F-FDG PET imaging in this study shows effects of sgACC/25 over-activation on both amygdala and hypothalamic activity. We have also previously shown links between sgACC/25 over-activity and reduced activity in the nucleus of the solitary tract[24]. Given the relationship between vagal tone and sgACC/25 activity demonstrated here, one might speculate that vagus nerve stimulation—a treatment for depression—may exert some of its therapeutic effects through modulation of sgACC/25. Indeed, chronic stimulation of the

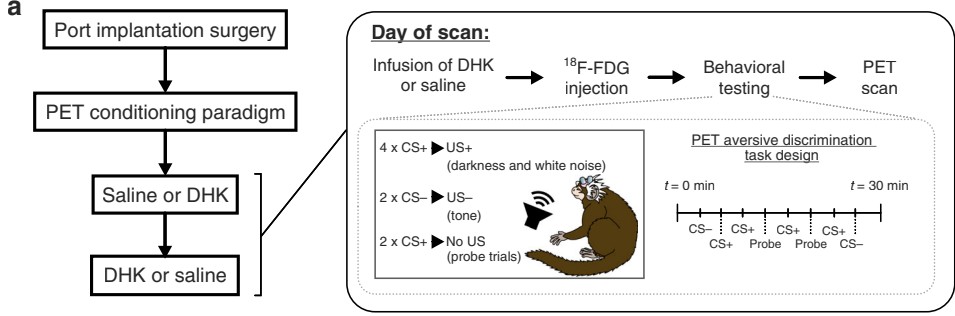

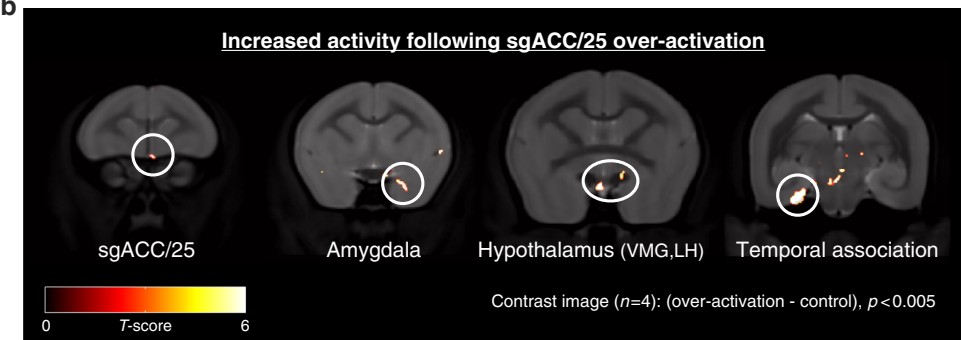

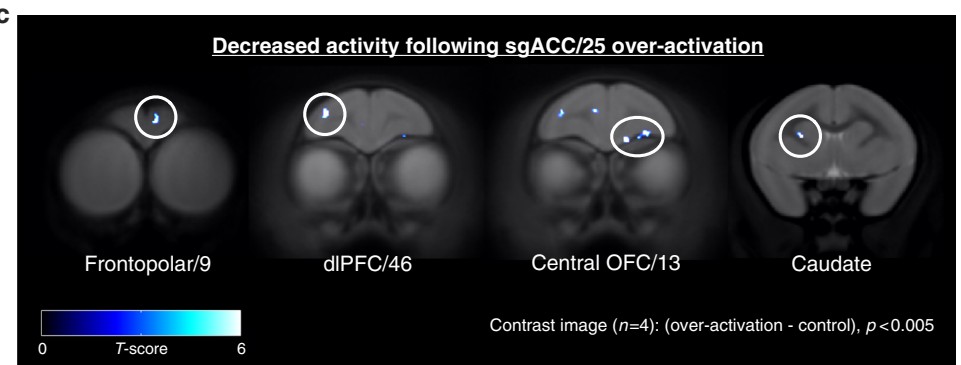

**Fig. 5 Circuit-wide changes associated with sgACC/25 over-activation revealed with $^{18}$F-FDG PET imaging. a** Four marmosets had a subcutaneous port implanted into the jugular vein and were trained on a lengthened version of the aversive Pavlovian discriminative conditioning test in preparation for scanning. Saline (control) and DHK (over-activation) scans were counterbalanced. On the day of the scan (inset), animals were infused with saline or DHK and immediately injected with $^{18}$F-FDG ligand through the port. The conditioning session lasted 30 min to increase sensitivity of the ligand to perturbations in brain activity caused by the paradigm. On the day of scans, this consisted of four CS+/US+ pairings, two CS−/US− pairings and two "probe" trials of a CS+ alone to limit aversive US exposures. These were presented in a fixed order indicated on the timeline. **b** Contrast images calculated from standardized uptake value ratio (SUVR) maps of over-activation—control scans, showing brain regions with increased activity following over-activation. These included sgACC/25, the amygdala, the ventromedial (VMH) and lateral (LH) hypothalamic nuclei and temporal association area TH ($p < 0.005$, uncorrected). **c** Contrast images calculated from SUVR maps of control—over-activation scans, showing brain regions with reduced activity following over-activation. These included frontopolar cortex/9, dorsolateral prefrontal cortex (dlPFC)/46, central orbitofrontal cortex (OFC)/13 and the lateral caudate ($p < 0.005$, uncorrected). Source data are available from the corresponding author upon request.

vagal nerve reduces subgenual prefrontal metabolism with the earliest changes detectable in sgACC/25, gradually extending rostrally to encompass vmPFC more broadly (including areas 10/14) over a 6–12-month-period[35] possibly reflecting the normalization of activity in a hyperactive sgACC/25.

Across two different Pavlovian conditioning paradigms of proximal threat, sgACC/25 over-activation enhanced cardiovascular and behavioral arousal in a manner unrestricted to CS periods, instead increasing arousal across the entire test session consistent with generalization to the context. Such effects mirror the elevated sgACC activity associated with contextual conditioning and during sustained or unpredictable threat in humans[18,19,36]. Both discrete and contextual Pavlovian processes are thought to contribute to anxiety disorders[37], but contextual

arousal is particularly relevant to psychopathologies characterized by sustained or "free-floating" anxiety when there is no clear threat-eliciting stimulus, such as generalized anxiety disorder. The generalized elevation in arousal associated with sgACC/25 over-activation may be consistent with the exaggerated responses to ineffable threat seen in these disorders, but contrasts with the view that the vmPFC acts to inhibit emotion-generating structures[38–40].

$^{18}$F-FDG PET imaging in the context of aversive Pavlovian discriminative conditioning revealed a somewhat different network of brain regions affected by sgACC/25 over-activity to those previously shown to be affected in the context of appetitive arousal[24]. While caution is warranted owing to the relatively small numbers of subjects inherent in pharmacological

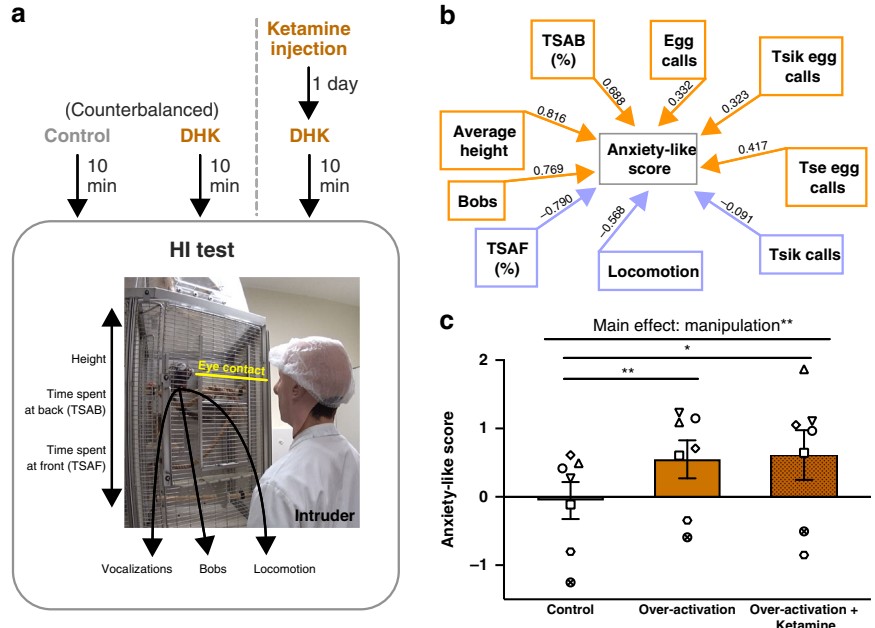

**Fig. 6 Ketamine does not reverse heightened anxiety-like behavior induced by sgACC/25 over-activation on the Human Intruder (HI) test.** Gray = control, orange = over-activation. Error bars represent SEM. $n = 7$. **a** All seven animals had three HI testing sessions. The first two sessions were counterbalanced, consisting of a control (saline) infusion or over-activation using DHK. The final session consisted of over-activation 24 h following ketamine administration. In the test, marmosets are divided into a quadrant of their cage and are confronted with a novel intruder who maintains eye contact for 2 min. Marmosets display a range of behaviors including vocalizations (tsik, tsik egg, tse egg, and egg calls), bobbing (rapid side-to-side movements of head and body), and locomotion. **b** These behaviors—together with average height, the time spent at the back of the cage (TSAB), and the time spent at the front of the cage (TSAF)—are loaded onto a single EFA-extracted factor representing anxiety-like behavior, with weightings shown above the arrows. **c** Replicating a previous finding from our group, sgACC/25 over-activation alone resulted in a significant increase in anxiety-like factor scores (one-way repeated measures ANOVA, effect of manipulation, $F_{1.6,9.7} = 13.3$, $p = 0.002$, $\eta^2 = 0.138$; control vs. over-activation, $p = 0.003$, $d = 2.23$). This effect was not reversed by ketamine (Sidak's test, over-activation vs. over-activation + ketamine, $p = 0.973$): compared to control conditions, sgACC/25 over-activation following ketamine administration still resulted in elevated factor scores (Sidak's test, control vs. over-activation + ketamine, $p = 0.020$, $d = 1.54$). Source data are provided as a Source Data file.

manipulation studies in NHPs, this exploratory analysis showed increased activity within the amygdala, hypothalamus and temporal lobe—regions known to be connected to sgACC/25[28,41]. Activity increases in the former two are consistent with the potentiation of a behavioral and cardiovascular stress response[42] while increases in the latter may be associated with contextual processing associated with the generalized elevation of arousal to the context. Reductions in activity were observed in frontopolar cortex, central OFC/13 and dlPFC/46. Reduced activity within dlPFC is consistent with neuroimaging findings in people with major depressive disorder, showing hypoactivity in dorsal PFC regions coupled with hyperactivity in ventral regions including sgACC[4,43]. dlPFC is recruited during the reappraisal of emotions, and its hypoactivity may thus reflect impaired regulation of emotion[44]. Reduction in its activity following sgACC/25 over-activation observed here could suggest that hyperactivity in more ventral regions may be causal in the disruption of higher-order emotion regulation structures, and that dorsal regions may need to undergo plastic changes over time in order to adapt to prolonged hyperactivity in sgACC/25.

In marked contrast to the comparability of these findings to studies in depressed and anxious humans, they appear in opposition to those pertaining to the putative rodent homolog—infralimbic cortex (IL)—during extinction. Electrophysiological and immunohistochemical evidence suggests that increased activity within IL correlates with reduced fear during extinction retrieval[45–47]. Pharmacological[48] and optogenetic[49] over-activation of IL also enhances extinction learning. Rodent studies, therefore, suggest that increased IL activity is associated with

reduced threat-elicited responding. The results reported here indicate the opposite in the case of the primate anatomical homolog: increased activity in marmoset sgACC/25 enhances threat-elicited responding. This calls into question the hypothesis that sgACC/25 and IL are functionally analogous and at the very least encourages a more in-depth investigation into the contextual variables across marmoset and rodent studies that could be responsible for such opposing effects.

Given our previous findings that ketamine successfully reverses sgACC/25 over-activation induced blunting of appetitive anticipatory arousal[24], it was important to determine whether ketamine could similarly reverse the over-activation induced enhancement of negative affect. A single i.m. dose of ketamine failed to ameliorate the heightened anxiety-like behavior in response to a human intruder. The enhanced responsivity to uncertain threat following sgACC/25 over-activation replicates previous work in our laboratory[24] and is consistent with findings in macaques showing that sgACC/25 activity is related to cortisol responses measured during intruder exposure[50]. Its unresponsiveness to ketamine would suggest that there are distinct downstream mechanisms at play underlying the enhancement in threat processing vs. blunted reward processing, consistent with the different patterns of network activity engaged following sgACC/25 activation in response to reward and threat. While ketamine's beneficial effects in people with depression and anhedonia are thought to depend on modulating sgACC activity[51], there are no imaging studies exploring the brain regions affected by ketamine in anxiety disorders. Comparing the neurobiological correlates of ketamine's action in these two conditions may yet provide further insight into its action.

This work lays the fundamental groundwork for several exciting future avenues exploring prefrontal contributions to stress-related disorders. The manipulations described in this study were acute whereas depression and anxiety are chronic conditions, so the effects of sustained sgACC/25 over-activity should be ascertained by using, for example, viral-mediated technologies. Such techniques could also fractionate the specific pathways that may underlie the blunted appetitive and heightened aversive responses induced by over-activation, which can co-exist in people with depression. Importantly, these techniques could also be used to identify the nature of the hierarchical interactions between higher-order prefrontal areas and sgACC/25 in the overall control of reward and threat-related behaviors.

The data presented here causally implicate over-activity in sgACC/25 in core features of stress-related disorders: both physiological dysfunction and enhanced cardiovascular and behavioral arousal in aversive contexts. While the vmPFC is often framed as an inhibitory regulator of subcortical emotion-generating structures, these data specifically implicate sgACC/25 over-activity as promoting a maladaptive stress response. The lack of efficacy of ketamine in reversing elevations of aversive arousal contrasts with our previous report of its efficacy in reversing over-activation induced blunting of reward anticipation. Altogether, these findings reveal the striking opponency of sgACC/25 over-activity on threat-elicited responses compared to reward responses, which parallel the anxiety and anhedonia-related symptoms of stress-related disorders, and their differential sensitivity to antidepressant treatment.

## Methods

**Subjects**. Seven marmosets (*Callithrix jacchus*, three females and four males) bred on-site at the University of Cambridge Marmoset Breeding Colony were housed in male/female pairs (males were vasectomized). All were experimentally naive at the start of testing.

Animals were kept in a 12-h light–dark cycle (lights on at 7 a.m., lights off at 7 p.m.) in a controlled environment of $22 \pm 1\,°C$ temperature and $50 \pm 1\%$ humidity. Their cages (2.80 x 1.20 x 0.98 m) contained a nest-box together with a variety of environmental enrichment aids, including suspended ladders, wooden branches, ropes, and boxes. Animals had ad libitum access to water and were fed a varied diet of MP.E1 primate diet (Special Diet Services, Essex, UK), carrot, fruit, rusk, malt loaf, eggs and bread. All procedures were carried out in accordance with the UK Animals (Scientific Procedures) Act 1986 and the University of Cambridge Animal Welfare and Ethical Review Body.

**Surgical procedures**. All animals underwent three aseptic surgical procedures: one to implant a telemetric blood pressure monitor into the descending aorta, a second to implant intracerebral cannulae targeting sgACC/25 and a third to implant a vascular access port for administration of $^{18}$F-FDG. Animals had 7–10 days of recovery following all surgical procedures.

**Telemetry surgery**. Marmosets were pre-medicated with ketamine (Ketavet; 10 mg/0.1 ml i.m. injection; Henry Schein, Melville, NY) before receiving the analgesic Carprofen (1.5 mg/0.03 ml s.c.; Pfizer, Kent, UK). They were intubated and maintained on 2.0–2.5% isoflurane in 0.3 l/min $O_2$. Animals were monitored throughout using pulse-oximetry and capnography (Microcap Handheld Capnograph, Oridion Capnography Inc., MA, USA) as well as a rectal thermometer measuring core body temperature (TES-1319 K-type digital thermometer, TES Electrical Electronic Corp, Taipei, Taiwan). The descending aorta was visualized within the abdominal cavity and the probe catheter of a telemetric blood pressure transmitter (HDS-10, Data Sciences International, USA) was implanted into the aorta just above the aortic bifurcation. Blood flow was occluded for no more than 3 min. The probe was sutured in place within the abdomen. All monkeys received meloxicam (0.15 mg/0.1 ml p.o.; Boehringer Ingelheim) postoperatively for three days in addition to prophylactic antibiotic treatment with amoxicillin and clavulanic acid (Synulox; 50 mg/ml solution, Pfizer, Kent, UK) one day before and for six days after surgery.

**Cannulation surgery**. Animals were anesthetized as before and placed into a stereotaxic frame modified for the marmoset (David Kopf, Tujunga, CA, USA). Cannulae (Plastics One, Roanoke, VA, USA) were implanted into sgACC/25 (26-gauge double cannulae, 7.0 mm protrusion from the base of the guide, 1.0 to 1.4 mm apart, anteroposterior [AP] + 12.5–13.8, lateromedial [LM] ± 0.7) adjusted in-situ according to cortical depth[52]. Postoperatively, animals received the analgesic

meloxicam as above. The cannulae and mount were cleaned weekly (and caps and cannula blockers changed) to ensure the cannulae were patent and the site remained free from infection.

**Port implantation surgery**. A vascular access port (Solomon Scientific, Skokie, IL, USA) was implanted into seven animals to allow swift subcutaneous injection of $^{18}$F-FDG. Animals were anesthetized and placed on to the surgical table in a prone position. An incision was made below the shoulder-blades perpendicular to the axis of the spine where the port would be placed, and a second incision was made on the neck to expose the jugular vein. A catheter attached to the port was threaded under the skin from the back towards the vein. The port was placed in a skin pocket created by the first incision. A small incision was made in the jugular vein to insert the open end of the catheter in the direction of the heart. The catheter was glued to the vein with Vetbond (3M Animal Care Products, St. Paul, MN, USA) and the incisions on the back and neck were sutured. All animals received meloxicam and amoxicillin/clavulanic acid perioperatively as above.

**Behavioral testing apparatus and paradigms**. In all paradigms except for the human intruder test, animals were trained to enter a transparent Perspex carry-box ($240 \times 230 \times 200$ mm) in which they were transported to the behavioral testing apparatus. The carry-box had two circular windows (diameter 30 mm) on opposite sides. The Perspex carry-box was placed inside the test chamber, and the marmoset remained inside this box during testing.

For practical reasons regarding constraints on the testing apparatus, the behavioral order of the tests was not counterbalanced: all animals progressed from testing in the neutral condition, to the conditioned threat and extinction paradigm, to the aversive Pavlovian discrimination test with the human intruder test done in parallel. For each behavioral paradigm, a within-subject control infusion was always carried out and in all cases during control infusions the animals showed the appropriate behavioral and cardiovascular responses to be expected in each paradigm.

**Neutral condition**. In the neutral condition, marmosets were placed inside a bespoke sound-attenuated testing chamber with the houselight turned on. Initial sessions were 5 min long, with the length extended to a maximum of 20 min over a period of 5–7 days. The animals were judged to have habituated when heart rate reached a stable level across two days (mean heart rate within ± 10%) and when animals showed a relaxed, still and non-vigilant posture. The total number of habituation sessions depended on the individual animal's rate of acclimatization to the apparatus (mean ± SEM number of sessions: 25 ± 6).

**Pavlovian conditioned threat and extinction test**. Pavlovian conditioned threat and extinction testing occurred inside a bespoke testing chamber. One wall of the chamber consisted of a pane of switchable SmartGlass (SmartGlass International®, Dublin, Ireland). The opacity of the SmartGlass pane is altered when voltage is applied, changing from opaque to transparent. When the SmartGlass is transparent, it reveals an additional section of the testing chamber. A rubber snake (US+) was placed in this chamber for specific trials of the acquisition session only.

A single block of the conditioned threat and extinction paradigm consisted of five sessions, run over five consecutive days (Supplement Fig. S1):

- Sessions 1–2 (habituation)—Animals were habituated to 12 trials (inter-trial interval (ITI) 110–130 s) of the SmartGlass illuminating for 5 s to show the empty additional chamber (unconditioned stimulus -; US-).
- Session 3 (acquisition)—Conditioned stimuli (CS) were introduced as 15 s, 70 dB auditory cues. Each of the nine trials consisted of a 15 s CS and a 5 s US (ITI 160–180 s). CSs persisted for the 5 s of the US to co-terminate with the US. The first three CS presentations were paired with the US-. The final six CSs were paired with the US+ (pane turning transparent to reveal the rubber snake in the additional chamber).
- Session 4 (extinction)—Each of the ten trials consisted of the 15 s CS followed by the US- (ITI 60–80 s). Control or over-activation infusions were carried out immediately prior to the extinction session to determine the effects of sgACC/25 activation on the expression and extinction of conditioned threat.
- Session 5 (extinction recall)—Ten CS/US- trials (ITI 70–110 s), identical to extinction, were presented to test for recall of threat extinction.

Three blocks of five sessions were repeated for each subject with a 1-week gap between blocks. These included two saline control blocks and one over-activation block. The first block (always saline) was used as a baseline block—the habituation sessions of this block were not comparable to the second and third blocks because the animal had never experienced the aversive snake in the chamber. The second and third blocks were used in data analysis, and control and over-activation manipulations were counterbalanced within these blocks. To minimize threat generalization between blocks, distinct patterned wall panels were introduced to vary the context and different auditory cues were used as CSs.

Four out of seven marmosets were tested on the Pavlovian conditioned threat and extinction test; three moved straight onto the aversive Pavlovian discriminative conditioning test following the neutral condition. Of these three, two marmosets' testing schedules were expedited due to SARS-CoV-2 while one marmoset failed to show sufficient reactivity to the snake US.

**Aversive Pavlovian discriminative conditioning test and PET scanning**. During the aversive Pavlovian discriminative conditioning test, seven marmosets were first exposed to two novel auditory cues (20 s) and the blood pressure response was measured. The cues were then counterbalanced such that the cue producing the smallest arousal response became the CS+ and the cue producing the largest arousal response became the CS−. The animals were then trained on an aversive Pavlovian discrimination paradigm (Supplement Fig. S4): the CS+ was associated with an aversive US+ consisting of 30 s of darkness, with 10 s of 85 dB white noise pseudo-randomly presented either in the first, second, or third 10 s of the darkness. The CS− was associated with a 0.5 s 80 dB neutral 2 kHz tone (US-). CSs co-terminated with their respective USs. ITIs were pseudo-randomly varied between 100–160 s. Each session consisted of two to four trials with no more than one CS+/US+ trial in a single session. Infusions were always conducted on sessions with a trial structure of CS−/CS+/CS−.

Seven marmosets were implanted with a subcutaneous port for PET scanning, but only four were scanned owing to one telemetry probe failure, one port blockage and one marmoset's scans halted due to SARS-CoV-2. The four marmosets undergoing $^{18}$F-FDG PET scanning were tested on a lengthened version of the aversive Pavlovian discrimination test. The length of the session was increased from 500 s to 1800 s to facilitate perturbation of cerebral $^{18}$F-FDG uptake by the behavioral paradigm. Habituation to these extended testing sessions took place over ten sessions using the same trial structure as above except with lengthened ITIs (330–440 s). During training, marmosets were also habituated to all handling procedures involved on a scan day, including mock injections and infusions but excluding anesthesia induction. Testing on this lengthened aversive discrimination test took place every day up until the final scan (except on recovery days following port implantation surgery or scanning).

The behavioral test on PET scan days was a unique version of the lengthened conditioning test, consisting of three trial types during the 30-min session: two CS−/US- trials, four CS+/US+ trials and two probe trials. The latter consisted of the standard 20 s CS+ but no US+, to limit aversive exposures in one testing session. The order of CS presentations is detailed in Fig. 5a, with ITIs of 130–180 s.

**Human intruder (HI) test**. The HI test was carried out in the marmosets' home cage. During testing sessions, animals were divided in the top right quadrant of the cage away from their cage mate. After 8 min of habituation to separation, an unfamiliar intruder entered the room. The intruder wore different latex masks to disguise their face. The intruder stood 40 cm from the front of the cage and maintained eye contact with the marmoset for 2 min. Behaviors were recorded using a video camera (GoPro 5, USA) and microphone (Sennheiser MKE 400, Germany). Following intrusion, animals were recorded for a further 5 min. The order of latex masks was counterbalanced and there was an interval of at least 1 week between each session.

**Drug treatments**. For all sterile drug treatments, marmosets were held gently by an assistant familiar to the animal while the researcher administered the drug. For central infusions, the caps and cannula blockers were removed from the guide, and the site was cleaned with 70% isopropyl alcohol. A sterile injector (Plastics One, Roanoke, VA, USA) connected to a 2 µl gas tight syringe in a syringe pump was inserted into the guide. The length of the injector was determined by cortical depth and the placement of the guide cannulae during surgery. Bilateral infusions were carried out over 2 min at a rate of 0.5 µl/min, with the injector left in place for a further minute to allow the drug to diffuse before injector removal. Sterile cannula blockers and caps were replaced, and the marmoset was returned to its cage for 10 min. For intramuscular injections of ketamine, the injection site located on the lateral thigh was cleaned with alcohol and injected with 0.5 mg/kg of ketamine or an equal volume of saline vehicle before testing. See Table S2 for a summary of timings and doses of the drugs used in this manuscript.

**PET imaging**. Each marmoset undergoing PET scanning received two $^{18}$F-FDG PET scans with a microPET Focus-220 scanner (Concorde Microsystems, Knoxville, TN, USA) with the first scan ~2 weeks after port implant surgery and an interval between scans of ~2 weeks. On the day of a scan, animals did not receive breakfast to lower blood glucose concentration and hence increase the transport of $^{18}$F-FDG into brain tissue, thereby increasing the cerebral $^{18}$F-FDG signal to noise ratio. The marmoset received an infusion of either saline vehicle or DHK 10 min prior to a bolus injection of ~70 MBq of $^{18}$F-FDG administered subcutaneously via the vascular access port. They were then taken to the test apparatus, and after the behavioral paradigm described above, the marmoset was anesthetized. The marmoset was then placed on a heat-pad on the scanner bed and attached to monitoring equipment. Heart rates, $SpO_2$ and respiratory rates were monitored continuously. The bed of the scanner was positioned to locate the brain in the center of the PET scanner field-of-view, where both sensitivity and resolution are optimal. For consistency, PET data acquisition started 70 min after the $^{18}$F-FDG injection and lasted for 45 min. The energy and coincidence timing windows used were 350–650 keV and 6 ns, respectively.

The list mode PET data were histogrammed into $9 \times 5$ min four-dimensional sinograms and then reconstructed using Fourier rebinning (FORE[53]), followed by the two-dimensional ordered subsets expectation maximization (OSEM) algorithm

installed on the scanner (6 iterations, 16 subsets). As post-injection transmission scanning was not feasible, attenuation correction used a mean non-attenuation corrected $^{18}$F-FDG image to determine a body outline, within which a uniform attenuation coefficient (0.096 cm$^{-1}$) was ascribed. This was combined with a standard attenuation map of the carbon fiber bed determined from transmission scanning. The combined attenuation map was forward projected using software installed on the scanner to produce an attenuation correction factor sinogram, and image reconstruction was repeated with attenuation correction applied. Corrections were also applied for random coincidences, dead-time, normalization, scatter, sensitivity and decay.

**Salivary cortisol sampling**. Salivary cortisol samples were collected in the affectively neutral condition (to measure basal cortisol levels) and in the acquisition and extinction sessions of the Pavlovian conditioned threat and extinction test (to measure cortisol dynamics during aversive conditions). A salivary sample of cortisol was taken during the infusion as a "pre" sample before the testing session, with a second "post" sample taken after testing. Salivary samples were taken by placing a sterile cotton bud into the marmoset's mouth and rotating it for 5–15 s to absorb sufficient saliva. Samples were then placed in sterile tubes and stored in a −20 °C freezer until processing. The samples were analyzed externally by immunoassay at the Core Biochemical Assay Laboratory (CBAL) within Addenbrookes' Hospital (Cambridge, UK). Salivary cortisol samples were taken at the same time of day for all animals (~11 and 11:30am) to control for cortisol's circadian rhythm[54]. In the neutral condition, four marmosets had cortisol levels measured but one sample had insufficient saliva, resulting in a sample size of three (Fig. 2h). In the conditioned threat and extinction paradigm, all four marmosets had salivary cortisol measurements during acquisition and extinction.

**Post-mortem histological processing**. Animals were pre-medicated with ketamine hydrochloride before being euthanized with sodium pentobarbital (20 mg/1 ml; Merial Animal Health, Essex, UK). Animals were then perfused transcardially with 300 ml 0.1 M PBS, followed by 300 ml of 4% paraformaldehyde fixative solution over ~15 min. The brain was removed and left in 4% paraformaldehyde fixative solution overnight before being transferred to 0.01 M PBS-azide solution for at least 48 h. Finally, the brains were transferred to 30% sucrose solution for at least 48 h for cryoprotection. Brains were then sectioned on a freezing microtome (coronal sections of 40–60 µm thickness), mounted on slides and stained with cresyl-violet. The sections were viewed under a light microscope (Leitz DMRD, Leica Microsystems, Germany). The cannula locations for each animal were schematized onto drawings of standard marmoset brain coronal sections (Fig. 1e).

**Data acquisition and statistical analysis**. For analysis, all data were inputted into GraphPad Prism 8 (GraphPad Software, La Jolla, CA, USA), IBM SPSS Statistics v25.0 for Macintosh (IBM, USA) and Microsoft Excel v16.37 for Macintosh (Microsoft, USA). Significance was set at $\alpha = 0.05$ in all cases. In the case of all analysis of variances (ANOVAs), multiple comparisons were calculated using Sidak's multiple comparisons test. Effect sizes are presented as Cohen's $d$ for $t$-tests, post-hoc tests and linear models, or $\eta^2$ for ANOVAs. * indicates $p < 0.05$, ** indicates $p < 0.01$ and *** indicates $p < 0.001$ in all figures.

Telemetry data collection and preliminary analysis: Blood pressure data were continuously transmitted by the implanted telemetry probe to a receiver for offline analysis using Spike2 (Version 8.11a, CED, Cambridge, UK). Any outliers and recording failures in the data were removed (values above 200 mmHg or below 0 mmHg, or other abnormal spikes). Data collection was reliable overall, but data gaps of less than 0.4 s were replaced by cubic spline interpolation and gaps of more than 0.4 s were treated as missing values. Systolic blood pressure (sBP) events were extracted as local maxima from each cardiac cycle, and diastolic blood pressure (dBP) events extracted as local minima. The heart rate was calculated using the time interval between adjacent maxima. MAP was calculated from adjacent systolic and diastolic values using the formula MAP = dBP + 1/3(sBP − dBP).

Neutral condition: In the neutral condition, blood pressure, heart rate and heart rate variability (HRV) data were collected. For infusion sessions, data were analyzed over minutes 1–10 where all animals appeared calm from behavioral and cardiovascular readouts. The 0th minute was excluded to allow time for acclimatization to the apparatus after transport. The main cardiovascular measures were blood pressure, heart rate and HRV. HRV was quantified by the root mean square differences of successive R-R intervals (RMSSD). Indices of sympathetic and parasympathetic activity, termed the cardiac sympathetic index (CSI) and cardiac vagal index (CVI), were also calculated as described in ref. [55].

sgACC/25 over-activation was compared to infusions of saline control on individual cardiovascular measures using two-tailed paired $t$-tests. Second-by-second heart rate values were compared using a linear mixed-model with time (in one-second bins) and manipulation (control, over-activation) as factors.

For all subsequent tests, blood pressure was used as the principal cardiovascular measure for two reasons: first, in both Pavlovian conditioning tests, blood pressure conditioning was more consistent compared to heart rate. Second, blood pressure was unaffected by sgACC/25 over-activation in the affectively neutral condition, whereas any changes in heart rate values would be confounded by a baseline

cardiovascular effect. The only case where heart rate values were used was in the PET conditioning paradigm, where responses were more consistent than blood pressure.

Pavlovian conditioned threat and extinction test: Behavior during CS periods was scored offline from video recordings of the session. The behavior scored was vigilant scanning (VS)—attentive scanning of the surroundings accompanied by a tense body posture[17,56,57]. VS measures were averaged across CS pairs (as described in ref. [58]). CS directed VS ($VS_{CS\ period}$ – $VS_{baseline}$, where baseline is 15 s pre-CS) was also calculated, which did show evidence of CS-specific learning (Supplement Fig. S2C). However, for consistency with cardiovascular measures (see below), absolute VS values were used for analysis.

Mean blood pressure values were calculated and averaged in CS pairs as above. Blood pressure responses were normalized (using a difference measure) to the mean blood pressure response across the two habituation sessions within the block (accounting for arousal to the context and individual differences in absolute blood pressure). CS directed blood pressure responses were also calculated and averaged across CS pairs, but consistent CS directed blood pressure responses were not evident (Supplement Fig. S2D) so absolute blood pressure values were used for analysis.

To determine if there was any difference in acquisition profiles during the "to be control" and "to be over-activation" blocks, the absolute blood pressure and absolute VS values of the final pre-acquisition CS pair were compared to the final post-acquisition CS pair across both blocks using a two-way repeated measures ANOVA of the form $M_2 \times P_2$: $M$ is a factor with two levels (manipulation) and $P$ is a factor with two levels (CS pair; pre- vs. post-acquisition). "Post":"pre" salivary cortisol ratios were also calculated for each acquisition block (four subjects, two blocks each for a total of eight blocks). These ratios were compared to a hypothetical value of 1.0 (no change) using a one-sample $t$-test, to determine if acquisition significantly elevated salivary cortisol.

Absolute VS and blood pressure values during the CS pairs of the extinction and extinction recall phases were analyzed using separate two-way repeated measures ANOVAs of the form $M_2 \times P_5$, where $M$ is a factor with two levels (manipulation) and $P$ is a factor with five levels (CS pair). Further analysis sought to determine whether there was a generalized effect of sgACC/25 over-activation on blood pressure or VS values during extinction/extinction recall sessions. For blood pressure, this was done by comparing blood pressure profiles across the entire extinction/extinction recall session (excluding the first minute) using a linear mixed-model with time (in one-second bins) and manipulation (control/over-activation) as factors. For VS, this was done by measuring baseline (15 s pre-CS) VS values across extinction/extinction recall sessions and comparing values between control and over-activation conditions using a two-tailed paired $t$-test.

Aversive Pavlovian discriminative conditioning test: The behavior scored was VS during the CS and baseline periods, and CS directed VS values were calculated as CS minus baseline as above. The cardiovascular readouts were mean blood pressure values during the 20 s baseline, 20 s CS and 30 s US+ periods. Both CS directed and US directed ($MAP_{US\ period}$ – $MAP_{CS\ period}$) blood pressure responses were calculated. Unlike the conditioned threat and extinction test, CS directed conditioning was evident for both VS and blood pressure in this paradigm and these measures were used in analysis. The 10 s period following termination of the US+ was defined as the recovery period—this period is of a priori interest as impaired stress recovery is a key feature of psychiatric disorders[59].

To illustrate successful discrimination between CS+ and CS−, a one-way repeated measures ANOVA was carried out comparing cardiovascular and behavioral responses to each trial on the CS−/CS+/CS− session immediately prior to infusions. To demonstrate a response to the US+, US directed blood pressure responses were compared to a hypothetical value of 0 (no change compared to CS period) using a one-sample $t$-test.

For drug manipulations, it was first determined if there were any differences in CS directed VS and blood pressure responses using a two-way repeated measures ANOVA of the form $M_2 \times C_3$, where $M$ is a factor with two levels (manipulation) and $C$ is a factor with three levels (CS type: CS-$_1$, CS+ or CS-$_2$). Further analysis sought to determine whether there was a generalized effect of sgACC/25 over-activation on absolute VS or blood pressure values during the baseline period. Absolute VS and blood pressure values were compared across infusion type using separate two-way repeated measures ANOVAs of the form $M_2 \times B_3$ where $M$ is a factor with two levels (manipulation type) and $B$ is a factor with three levels (baseline type: first, second, or third).

During the 10 s recovery period after the US+, ratios were calculated comparing the blood pressure value in each second to the blood pressure value in the final second of the US+ period. Ratio values for control and over-activation conditions were compared using a two-way repeated measures ANOVA of the form $M_2 \times T_{10}$ where $M$ is a factor with two levels (manipulation type) and $T$ is a factor with ten levels (ten 1 s time bins).

For the modified discriminative conditioning for PET scanning, cardiovascular, and behavioral variables were analyzed across CS, US, and baseline periods for all trials similar to the standard aversive discrimination test. Mean CS directed data were analyzed using a two-way repeated measures ANOVA of the form $M_2 \times C_3$, where $M$ is a factor with two levels (manipulation type) and $C$ is a factor with three levels (CS type: CS−, CS+ and probe).

Human intruder (HI) test: A behavioral analysis program (JWatcher v1.0, UCLA and Macquarie University) was used to score behavior during the intruder phase. The measures scored were: time spent at the front of the cage (TSAF), time

spent at the back of the cage (TSAB), average height, average depth, bobbing (rapid side-to-side body movements), jumps, locomotion and vocalizations (egg calls, tsik call, tsik egg calls, and tse egg calls). Using exploratory factor analysis (EFA) with a principal axis factoring extraction method, the behaviors were loaded onto a single latent factor representing anxiety-like behavior (with weightings shown in Fig. 5b; see ref. [29] for detailed description). Once the EFA-derived anxiety scores were calculated for each HI session, a one-way repeated measures ANOVA was carried out to compare the effects of control, over-activation and [over-activation + ketamine] manipulations on anxiety scores.

The effects of ketamine were only investigated on the HI test owing to limitations on the number of infusions each animal could receive and timing constraints on the overall duration of the study.

PET scanning: Magnetic resonance (MR) imaging of the animals was not possible due to the cannulae implanted in the brain, preventing the use of MR-based spatial normalization. Instead, first, the mean 18F-FDG PET image of each scan was manually, rigidly registered to an 18F-FDG PET brain template produced from another 18F-FDG PET study in the colony that included MR imaging. The 18F-FDG PET brain template was constructed by averaging $n = 21$ 18F-FDG PET images transformed to template space using registration transformations obtained by warping MR images (co-registered to the 18F-FDG PET images) to an MR template. Second, for each subject, the two 18F-FDG PET scans rigidly registered to the 18F-FDG PET template were averaged, the resultant image was non-rigidly registered (affine and non-linear) to the 18F-FDG PET template using ANTS[60], and this transformation was applied to each of the three rigidly registered 18F-FDG PET scans. Use of a single spatial normalization transformation per subject rather than per 18F-FDG PET scan was adopted after it was found— using the $n = 21$ 18F-FDG PET scans with MR—that this approach provided region of interest (ROI) PET values with a higher correlation to those obtained using MR-based spatial normalization ($R^2 = 0.89$ vs. $R^2 = 0.87$).

For each scan, an SUVR map was created for voxel-wise analysis by dividing the mean PET image by a cerebellum ROI value. Normalization by the cerebellum signal was designed to minimize the confounding influence of inter-scan differences in tracer availability; plasma glucose concentration; the effect of anesthesia on cerebral blood flow and metabolism; and basal cerebral glucose metabolism.

SPM8 (Wellcome Trust Institute for Neurology, UCL, UK) was used for voxel-wise analysis. A general linear model was configured with covariates for subject and condition (saline control vs. over-activation) and changes in activity were tested with Student's $t$-test at each voxel. Prior to estimating the model, images were smoothed with a filter size of 1 mm$^3$ using a locally adapted Gaussian kernel to include only those voxels inside a brain mask. With the low numbers of subjects inherent in invasive primate studies, it is difficult for findings to survive conservative corrections for multiple comparisons. To provide an exploratory analysis with some mitigation against type I errors, an adjusted uncorrected threshold of $p < 0.005$ was used in combination with a small extent threshold of five voxels to remove implausible clusters from the statistical maps.

**Reporting summary**. Further information on research design is available in the Nature Research Reporting Summary linked to this article.

## Data availability
Source data are provided with this paper using Mendeley Data (https://doi.org/10.17632/tvxf6f9gj6.1). Raw neuroimaging data are available from the corresponding author upon request.

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

## Acknowledgements

This research was funded by a Wellcome Trust Investigator award (108089/Z/15/Z) to A. C.R. L.A. was funded by a Medical Research Council studentship. H.F.C. was funded by a Medical Research Council Career Development Award (RG62920).

## Author contributions

Conceptualization, L.A., H.F.C., and A.C.R.; Methodology, L.A., P.L.R.G., H.F.C., and A.C.R.; Investigation, L.A., C.W., P.L.R.G., T.D.F., Y.T.H., L.M., and H.F.C.; Formal analysis, L.A., C.W., S.J.S., T.D.F., and Y.T.H.; Writing—original draft, L.A., C.W., and A.C.R.; Writing—review and editing, L.A., C.W., and A.C.R.; Visualization, L.A., C.W., S.J.S., T.D.F., and Y.T.H.; Funding acquisition, A.C.R.; Resources, A.C.R.; Supervision, A.C.R.

## Competing interests

The authors declare no competing interests.
