## [Peer Review File · Nature Communications]

Reviewers' Comments:

Reviewer #1:

Remarks to the Author:

This is an in-depth study of the role that a subregion of the marmoset monkey cingulate plays in negative emotional processing. A major strength of the study is its integrative nature. Using a fear conditioning approach and manipulation of neural activity, the authors measure behavioral outputs (visual scanning), heart rate and its variability, cortisol levels and PET imaging. The authors also use a ketamine treatment to see if it intervenes with processes elicited by the negative emotions. Overall, I think this is a very high quality study.

Here are my suggestions which I hope are helpful:

1. One of the justifications for using marmosets in this study is that, relative to rodents, their frontal cortical organization should be more similar to human frontal cortex. I think a more elaborated justification is needed, perhaps reiterating information from Roberts & Clarke (PNAS)(maybe even a version of Figure 1 from that paper). After reading the manuscript, I was left feeling that another reader might think all of this could've been done in rodents. More justification will also make the punch about rodent infralimbic cortex in the Discussion land better.
2. The authors focus on sgACC/25 and make very strong claims about this specific subregion. Those claims are strong enough that it seems that a control manipulation in adjacent cortical region may be warranted. Short of doing that, could the authors address this issue?
3. The authors might want to describe a little more in the Results the exact nature of the fear conditioning paradigm used here and what hypotheses are generated from it that they are explicitly testing. For example, I have a reasonably idea of what it is but mainly from LeDoux's work. For example, I do not understand what extinction recall is. I don't presume everyone will be as ignorant as me, but there might be more than a few.
4. For the aversive discrimination test, why were the US's so different in kind. What I mean, the duration of the US+ was 30 seconds while US- was only 500 milliseconds. Likely, I am misunderstanding something but it seems that there should be more equivalency. As it stands, the discrimination seems too easy to be an effective measure.
5. If the same marmosets are being used across different aspects of the study, how does "experience" potential effect interpretation of the Results?
6. The Human Intruder test seems to come out of nowhere. Why wasn't the same approach as the other parts of the study used to measure the putative effects of ketamine therapy?
7. Finally, the PET imaging data were a bit underwhelming. I'm not sure I learned very much from those data as presented in the current manuscript. What were the specific hypotheses tested and how are they supported by up versus down activity levels in various regions? And what about brainstem regions or regions not directly connected to ACC25?

signed

Asif Ghazanfar

Reviewer #2:

Remarks to the Author:

This is a strong, interesting and important study combining behavioral pharmacology to reveal the functional role of primate subgenual cingulate cortex in cardiac and cortisol responses to threat. Local infusion in increase activation within subgenual cingulate was associated with cardiovagal withdrawal increased cortisol release and increased cardiac responses to threat. FDG PET imaging showed association with amygdala, DLPFC and hypothalamic activation. Ketamine administration after threat learning did not reverse overactivation in threat responses. The paper provides important new data on the functional circuitry of mood and anxiety disorder and the link with stress and physiological arousal. In my comments, clarification is asked for about mechanism and anatomy of the effects.

General

The overactivation of subgenual cingulate neurons using DHK presumably disrupts integrative processing of information i.e. the region's increased neural excitability / metabolic activity is inefficient (as it might be in mood disorder), so that suppression ameliorates deleterious consequences. In this paper, increased activity occurs with cardiovagal withdrawal allowing sympathetic cardiac dominance. This effect contrasts with studies that implicate subgenual cingulate activity in enhancing cardiovagal tone, and baroreflex function showing negative relationships with both VMPFC and sbgenual cingulate with psychological and physiological arousal (e.g. Verberne et al 1987, Fryszak & Neafsey 1994, Rolls et al 2003, Nagai et al 2004, Gianaros and Wager 2015). A line or two about the likely circuit level mechanism of the overactivation would be helpful. This is also relevant to the discussion from 305 onwards.

The likely autonomic mechanisms involved could be clarified, as there seem to be more than one process. e.g. if heart rate not blood pressure is changed is this more likely to be cardiovagal withdrawal than increased sympathetic drive? Are the main autonomic effect via suppression of baroreflex sensitivity? The conditioned effects on bp, cortisol seem to be via sympathetic enhancement; is there a stronger link with vigilance? What is known about how subgenual cingulate directly and via amygdala or hypothalamus connects to autonomic brainstem relays (e.g. Chiba et al., 2001)?

Minor points

- 1) Ketamine is not usually classed as an antidepressant (abstract & line92) but as a dissociative anesthetic agent
- 2) Dissociation states occur as trauma and anxiety symptom set and theoretically connected to autonomic uncoupling from cognition. Is this work on overexcitation of subgenual cingulate (or predicted effect of ketamine) relevant to this?
- 3) Line 139 implies predation by rubber snakes.
- 4) In PET study, do 2 activations occur in area 13? Is lateral OFC is the right term as area 13 can be described as mid OFC with orbital 12 being lateral?
- 5) Line 259 I am unsure what the predictions are about vagus nerve stimulation: if VNS stimulates visceral afferents from barorec

Reviewer #3:

Remarks to the Author:

This paper describes behavioral, autonomic, psychophysiological, and metabolic readouts of overactivation of the subgenual anterior cingulate cortex (sgACC). The contribution of this region to disorders of negative emotion, prototypically depression and anxiety, have been well described in

humans, with link of SCC overactivity to both high negative affect (sadness, anxiety) and low positive affect (anhedonia), including models defined in rodent sgACC analogues. This paper extends previous work from this group using a novel pharmacological overactivation model in marmosets to focus attention on enhanced negative –using a well-established threat model. This experiment demonstrates that overactivation of the sgACC enhances stress-induced threat responses with associated changes in vagal tone and cortisol release. Importantly, PET metabolic maps are distinct from the anhedonia induction demonstrated in previous experiments, confirming distinct sgACC networks mediating different negative states. Interestingly, enhanced negative as measured in this anxiety model is not affected by ketamine in contrast to induced anhedonia.

The experiments in this paper are well described, the chosen readouts and statistics are appropriate to the questions posed; the discussion articulates the potential clinical significance of the findings and importantly the critical distinctions of these findings to those demonstrated in putatively comparable rodent models. The use of marmosets is an important development for animal models of human psychopathology as their brain anatomy has clear homologies to humans and larger NHPs unlike rodents. The methods are detailed adequately to be replicated by others, with a clear roadmap for additional studies.

These findings provide strong evidence that hyperactivation of an anatomically well defined subregion of the prefrontal cortex in NHPs mediates a well characterized behavioral state. Further, the study presents evidence that there is clear circuit overlap with previous human studies of related behaviors, with significant and interpretable differences from studies performed in rodents. Such findings have important implications for establishing tractable animal models of human emotional behavior and argue for increased use of marmoset for these types of studies. The findings also make a strong case for the need for clear network-wide distinctions between anhedonia (low positive) and enhanced negative (arousal, fear) in studies attempting to model human mood disorders. These findings should have influence on design and interpretation of future human and animal models of depression and anxiety and have clear implications for novel treatment development and studies of treatment mechanisms.

Specific questions

1. While the focus on the subgenual anterior cingulate is well argued, one might also have examined the pregenual anterior cingulate for these experiments, especially given the focus on threat, fear conditioning and extinction. Were injections in the pregenual cingulate (area 24, 32) tested as control regions? If not, please discuss as a limitation.
2. Given the autonomic effects demonstrated, it is surprising not to see any metabolic changes in the dorsal ACC/Mid-cingulate, midbrain/brainstem (parabrachial nucleus, PAG). There are multiple human studies that would implicate those regions in the autonomic behaviors evoked here that might be reviewed and or discussed (see papers by Critchley, Riva Posse, Strick, among others).
3. Given the focus of previous overactivation studies in the sgACC, do animals in the stress condition with sgACC overactivation demonstrate any of the anhedonic behaviors previously described? Are there methods to measure both within the same session? Humans can be both anxious and anhedonic; how might that be modelled using this probe?
4. Lateralized PET findings. Were injections in the SCC unilateral or bilateral? The scans show clear laterality that is difficult to interpret (i.e., Left cortex/Right amygdala). Some discussion of laterality especially given the vagal findings seems warranted.

5. PET images are very small and hard to see; images might be larger so one can appreciate the anatomical location of the metabolic changes.

6. Might include discussion of other potential stress models that might address enhanced negative affect in depression more directly (in contrast to fear, threat, anxiety). Would be curious how a learned helplessness stress model or mild chronic stress would be impacted by sgACC overactivation.

7. As both depression and anxiety are chronic conditions; the discussion might expand on the utility of this model for inducing sustained state changes appropriate for understanding pervasive anhedonia and or enhanced negative affect seen in human disorders.

REVIEWER COMMENTS

Reviewer #1 (Remarks to the Author):

1. One of the justifications for using marmosets in this study is that, relative to rodents, their frontal cortical organization should be more similar to human frontal cortex. I think a more elaborated justification is needed, perhaps reiterating information from Roberts & Clarke (PNAS) (maybe even a version of Figure 1 from that paper). After reading the manuscript, I was left feeling that another reader might think all of this could've been done in rodents. More justification will also make the punch about rodent infralimbic cortex in the Discussion land better.

Thank you for this comment; we agree, the contrast with rodent IL could be made more evident. We have expanded on this in the introduction:

Studies in non-human primates (NHPs) are essential in addressing this question since, compared to rodents, the structural organization of the vmPFC in NHPs is far closer to that of humans. The differences between rodent and primate are highlighted by our recent findings that inactivation of marmoset sgACC/25 and pgACC/32 has opposing functional effects to that seen following inactivation of their putative rodent anatomical homologues (infralimbic cortex and prelimbic cortex respectively). Indeed, the wide and persistent translational gap between advances in preclinical research and a relative failure to develop more effective treatments for psychiatric disorders is due, in part, to a lack of understanding of the complex control of negative affect exerted by the highly evolved primate vmPFC.

2. The authors focus on sgACC/25 and make very strong claims about this specific subregion. Those claims are strong enough that it seems that a control manipulation in adjacent cortical region may be warranted. Short of doing that, could the authors address this issue?

We have already targeted adjacent pgACC/32 in two other published papers^{1,2} in which we demonstrate very clearly that manipulations in this region are completely different in all behavioural paradigms investigated (both reward and distal threat) to that seen following sgACC/25 manipulations. Thus, the selectivity of our sgACC/25 infusions has already been demonstrated. We now make this clear in the discussion:

Until now it was unknown whether altered vmPFC activity demonstrated in correlative neuroimaging studies of mood and anxiety disorders could cause the dysregulation of negative emotion apparent in these disorders, and if so, where the critical locus was located within vmPFC and the direction of its effects. The findings here directly address these questions by showing that over-activation of the caudal subgenual region of primate vmPFC, sgACC/25, reduces basal HRV and enhances both proximal and distal threat-evoked cardiovascular and behavioral arousal mirroring the changes observed in stress-related disorders. We have previously demonstrated the selective effects of sgACC/25 manipulations compared to adjacent brain regions, highlighting that this causal link is specific to sgACC/25.

3. The authors might want to describe a little more in the Results the exact nature of the fear conditioning paradigm used here and what hypotheses are generated from it that they are explicitly testing. For example, I have a reasonably idea of what it is but mainly from LeDoux's work. For example, I do not understand what extinction recall is. I don't presume everyone will be as ignorant as me, but there might be more than a few.

In the introduction, we have now elaborated on the different tasks used and the aspects of different threat processing they are measuring.

See below:

First, we ascertained the impact of sgACC/25 over-activation on cardiovascular and endocrine (cortisol) dynamics in an emotionally neutral condition. We then assessed the effect of sgACC/25 over-activation on several indices of affect regulation during both proximal and distal threat as defined by the predatory imminence theory, in which threat engages distinct cognitive and behavioral strategies depending on its closeness in time and space.

Proximal threat included (i) using a conditioning/extinction paradigm. The first test of the regulation of proximal threat involved a Pavlovian conditioned threat and extinction paradigm with a rubber snake as an ethologically relevant unconditioned stimulus (US). This paradigm was used to investigate the effects of sgACC/25 over-activation on the extinction of a conditioned threat response, and the subsequent recall of that extinction memory the next day. The second test focused on the ability of animals to discriminate between threatening and safety cues threat anticipation and stress recovery during an aversive Pavlovian discriminative conditioning paradigm. We utilized positron emission tomography (PET) imaging of the glucose analogue 2-deoxy-2-[¹⁸F]fluoro-D-glucose (¹⁸F-FDG) during this paradigm to determine the circuit-wide changes associated with sgACC/25 over-activation. Finally, we determined whether increased reactivity to distal threat induced by sgACC/25 over-activation on the human intruder (HI) test we looked to replicate our previous finding of sgACC/25 over-activation enhancing arousal to distal threat in the form of a human intruder. The novel question we addressed here was whether this effect could be ameliorated by ketamine, in the same way ketamine had ameliorated the blunting of appetitive anticipatory arousal we had demonstrated previously.

4. For the aversive discrimination test, why were the US's so different in kind. What I mean, the duration of the US+ was 30 seconds while US- was only 500 milliseconds. Likely, I am misunderstanding something but it seems that there should be more equivalency. As it stands, the discrimination seems too easy to be an effective measure.

The majority of threat conditioning tests in rodents only have a single CS and there is no discrimination. Where discriminations are used, very often there is no US-: the CS- is presented by itself. On this occasion we presented an actual US- because we know from animal learning studies that it is easier for animals to discriminate between CSs that are each followed by an event, i.e. US. The durations of the US+/- per se, do not impact on the difficulty of the CS discrimination.

It is also important to be aware that the main issue encountered when training animals on threat discriminations is not the risk of a discrimination being too simple; rather, it is that the discrimination is too difficult because there is a high risk of generalisation. In aversive situations there is a tendency for animals to associate the threat (rubber snake) with the entire test apparatus rather than with one specific cue in the test apparatus (known as generalisation). Making the USs distinct is critical in preventing this.

5. If the same marmosets are being used across different aspects of the study, how does "experience" potential effect interpretation of the Results?

We cannot address this issue of temporal order very easily. The behavioural order wasn't counterbalanced, but for each behavioural paradigm we always had a within-subject control infusion. In all cases, during control infusion sessions the animals showed the appropriate

behavioural and cardiovascular responses to be expected in each paradigm. We have made this clear in the methods under 'Behavioural testing apparatus and paradigms':

For practical reasons concerning constraints on the testing apparatus, the behavioral order of the tests was not counterbalanced: all animals progressed from testing in the neutral condition, to the conditioned threat and extinction paradigm, to the aversive Pavlovian discrimination test and finally to the human intruder test. For each behavioral paradigm, a within-subject control infusion was always carried out and in all cases during control infusions the animals showed the appropriate behavioral and cardiovascular responses to be expected in each paradigm.

6. The Human Intruder test seems to come out of nowhere. Why wasn't the same approach as the other parts of the study used to measure the putative effects of ketamine therapy?

There are two main reasons why we couldn't test the effect of ketamine on all of the paradigms described in this study: first, the number of brain infusions an animal can receive and second, the time taken for ketamine's effect to dissipate.

With regards to the first issue, it is impossible to carry out ketamine manipulations on every paradigm because of the number of infusions required. The study already required several infusions into sgACC/25. Ketamine studies require their own control infusions as well as over-activation infusions in the presence of ketamine, potentially doubling the total number of infusions needed.

Second, ketamine can have sustained effects that take three weeks to subside as we have shown previously². If ketamine manipulations were carried out on every paradigm, there would be a substantial amount of 'rest' time in between. The study already took between 12-16 months to perform with many different paradigms involved. We have added an explanation for this in methods under 'Data acquisition and statistical analysis':

The effects of ketamine were only investigated on the HI test owing to limitations on the number of infusions each animal could receive and timing constraints on the overall duration of the study.

7. Finally, the PET imaging data were a bit underwhelming. I'm not sure I learned very much from those data as presented in the current manuscript. What were the specific hypotheses tested and how are they supported by up versus down activity levels in various regions? And what about brainstem regions or regions not directly connected to ACC25?

We were disappointed that the reviewer was underwhelmed – equivalent neuroimaging data have never been seen before. Our finding that sgACC/25 overactivation reduced activity in dlPFC/area 46 and more dorsal cognitive regions, alongside reductions subcortically in limbic regions provides, for the first time, evidence of causality. So far, human studies have only shown correlations between the activity in these different brain regions in relation to negative affect and its regulation. Here, we are able to show that over-activation, selectively in sgACC/25, can cause not only upregulation in downstream subcortical limbic regions but reduced activation in higher-order cognitive areas, highly relevant to the cortico-limbic theory of brain dysregulation in depression³. We have tried to emphasise this more in the discussion. It must also be appreciated, however, that with the low numbers of animals in this study, that the power of this neuroimaging investigation is limited: we can only highlight the effects with big enough effect sizes despite the small number of subjects.

Reviewer #2 (Remarks to the Author):

General

1. The overactivation of subgenual cingulate neurons using DHK presumably disrupts integrative processing of information i.e. the region's increased neural excitability / metabolic activity is inefficient (as it might be in mood disorder), so that suppression ameliorates deleterious consequences. In this paper, increased activity occurs with cardiovagal withdrawal allowing sympathetic cardiac dominance. This effect contrasts with studies that implicate subgenual cingulate activity in enhancing cardiovagal tone, and baroreflex function showing negative relationships with both vmPFC and subgenual cingulate with psychological and physiological arousal (e.g. Verberne et al 1987, Fryszak & Neafsey 1994, Rolls et al 2003, Nagai et al 2004, Gianaros and Wager 2015). A line or two about the likely circuit level mechanism of the overactivation would be helpful. This is also relevant to the discussion from 305 onwards.

The perigenual and subgenual portions of the vmPFC have indeed been related to vagal reactivity previously. For example, variation in the high-frequency component of HRV (thought to reflect mainly parasympathetic tone) is strongly correlated with pg/sgACC BOLD signal^{4,5}. However, it is important to note that the areas of activation in these studies correspond to BA10, BA24 and BA32 – not BA25 which lies more caudally and outside of the region of activation seen in these studies.

One study that has implicated sgACC/25 specifically in autonomic reactivity is by Lane and colleagues⁶, which does show a correlation between sgACC/25 activity and vagal reactivity. However, this correlation is lost in the depressed state, suggesting an uncoupling of the vagal-prefrontal control mechanisms which could well be consistent with the results demonstrated here.

Consistent with our effects shown here, we have previously demonstrated that inactivation of sgACC/25 produces the opposite effects to those of over-activation demonstrated here¹. Thus, whereas inactivation of sgACC/25 increases vagal tone, over-activation decreases it. Stress-related disorders including depression are characterised by enhanced negative emotion together with reductions in heart rate variability. That sgACC/25 over-activity could be causally related to both of these features is a novel finding.

We have expanded on the likely circuit-level mechanisms of the autonomic effects in the discussion:

In affectively neutral conditions, sgACC/25 over-activation significantly increased basal heart rate and reduced HRV, an effect mediated by a shift in the parasympathetic-to-sympathetic balance similar to that apparent in mood and anxiety disorders. The perigenual and subgenual portions of the vmPFC have been related to vagal reactivity previously. For example, variation in the high-frequency component of HRV (thought to reflect mainly parasympathetic tone) is strongly correlated with pg/sgACC BOLD signal. However, it is important to note that the areas of activation in such studies typically correspond to areas 10, 24 and 32 – not area 25. We have shown here that there is a causal relationship between elevated sgACC/25 activity and reduced parasympathetic tone, consistent with previous findings from our group with respect to inactivation.

These results are also consistent with the anatomical connectivity of sgACC/25 identified from tracing studies. Neurons from primate sgACC/25 project directly to the brainstem but also indirectly via hypothalamic and amygdala nuclei indicating that sgACC/25 has dual access to an emotional-visceral motor system. Consistent with this, the ¹⁸F-FDG PET imaging in this study shows direct effects

of sgACC/25 over-activation on both amygdala and hypothalamic activity, and we have previously shown links between sgACC/25 over-activity and reduced activity in the nucleus of the solitary tract.

Furthermore, given the relationship between vagal tone and sgACC/25 activity demonstrated here, one might speculate that vagus nerve stimulation – a treatment for depression – may exert some of its therapeutic effects through modulation of sgACC/25. Indeed, chronic stimulation of the vagal nerve reduces subgenual prefrontal metabolism with the earliest changes detectable in sgACC/25, gradually extending rostrally to encompass vmPFC more broadly (including areas 10/14) over a 6-12 month period possibly reflecting the normalization of activity in a hyperactive sgACC/25.

2. The likely autonomic mechanisms involved could be clarified, as there seem to be more than one process. e.g. if heart rate not blood pressure is changed is this more likely to be cardiovagal withdrawal than increased sympathetic drive? Are the main autonomic effect via suppression of baroreflex sensitivity? The conditioned effects on bp, cortisol seem to be via sympathetic enhancement; is there a stronger link with vigilance? What is known about how subgenual cingulate directly and via amygdala or hypothalamus connects to autonomic brainstem relays (e.g. Chiba et al., 2001)?

Whilst not affecting baseline blood pressure, sgACC/25 over-activation significantly increased resting heart rate and reduced heart rate variability. By fractionating heart rate variability into sympathetic and vagal components, we show that the cardiovascular effects of sgACC/25 over-activation appeared to be mediated by both a reduction in CVI and increase in CSI. Nevertheless, a predominant influence of sgACC/25 on parasympathetic – rather than sympathetic – branches of the autonomic nervous system would explain why over-activation is associated with an increase in heart rate (under predominant parasympathetic control through vagal innervation of the sinoatrial node), without an effect on blood pressure (predominant sympathetic control through vasomotor actions at the arteriolar level) in ‘at rest’ conditions. This may be via an effect on baroreflex sensitivity – which would be consistent with some human work⁷ – although this was not measured in this study.

It is absolutely the case that vmPFC regions connect both directly and indirectly to brainstem regions involved in autonomic regulation. Neurons from non-human primate sgACC/25 have extensive projections to hypothalamic and amygdala nuclei which then project to spinal autonomic centres⁸⁻¹⁰. The vmPFC also directly innervates the brainstem, meaning it has dual access to an emotional-visceral motor system. We have expanded on this in the discussion: see response above.

Minor points

1. Ketamine is not usually classed as an antidepressant (abstract & line92) but as a dissociative anesthetic agent.

Ketamine has been approved as an antidepressant agent by the FDA (SPRAVATO[®]), and the doses we used in the study were consistent with its antidepressant – rather than anesthetic – effects.

2. Dissociation states occur as trauma and anxiety symptom set and theoretically connected to autonomic uncoupling from cognition. Is this work on overexcitation of subgenual cingulate (or predicted effect of ketamine) relevant to this?

This is an interesting question. We have previously shown that orbitofrontal lesions cause an uncoupling of the behavioural and autonomic features of appetitive Pavlovian conditioning

when CSs are presented in extinction, illustrating that dysfunction in adjacent prefrontal regions can mediate the dissociation of behaviour and autonomic function¹¹.

Whether something similar is true of over-activity in sgACC/25 – and whether this could be responsible for dissociative symptoms – remains to be seen. There are several studies which have shown that sgACC/25 is an important 'hub' in mediating coordination between cognitive and emotional networks, and that it may be involved in dissociative symptoms in some way, shape or form:

- sgACC/25 activity modulates connectivity between lateral (cognitive) and medial (affective) regions of the prefrontal cortex during normal sadness suggesting it is important in coordinating emotional-cognitive networks¹². One might extrapolate that dysfunctional activity in sgACC/25 associated with depression may result in a failure of these networks to coordinate their activity.
- Regions of the anterior cingulate (although not sgACC/25) respond more during emotional conflict in the Stroop task, with detectable changes in HRV and, in depressed patients, increased reports of dissociative symptoms¹³.
- Opioid-receptor binding in sgACC/25 increases in people with PTSD, a disorder characterised by dissociative experiences¹⁴.

Nevertheless, this discussion is highly speculative and too exploratory to include in the main manuscript.

3. Line 139 implies predation by rubber snakes.

This has been changed; thank you.

The sight of a rubber snake (~~a natural predator of marmosets and ethologically aversive~~) (snakes being a predator of marmosets and ethologically aversive) replaced aversive foot shock as the US (as in ¹) and was paired with a 15s neutral tone, the conditioned stimulus (CS; Supplement Fig. S2).

4. In PET study, do 2 activations occur in area 13? Is lateral OFC is the right term as area 13 can be described as mid OFC with orbital 12 being lateral?

There are indeed two foci, both within area 13.

We agree that BA13 is more central than lateral. We have changed the terminology of lateral OFC to central OFC.

5. Line 259 I am unsure what the predictions are about vagus nerve stimulation

We know that vagal nerve stimulation is correlated with reduced activity within sgACC/25 as measured by functional neuroimaging¹⁵. In this paper we demonstrate a direct relationship, albeit in the other direction, by revealing that sgACC/25 activation regulates heart rate variability and vagal nerve activity. We have made this more explicit in the manuscript.

Reviewer #3 (Remarks to the Author):

1. While the focus on the subgenual anterior cingulate is well argued, one might also have examined the pregenual anterior cingulate for these experiments, especially given the focus on threat, fear conditioning and extinction. Were injections in the pregenual cingulate (area 24, 32) tested as control regions? If not, please discuss as a limitation.

As discussed above in reply to Reviewer 1 (point 2), we have already targeted adjacent pgACC/32 in two other published papers^{1,2} where completely different effects were observed compared to sgACC/25. Studying pgACC/32 on these paradigms would require a new cohort of animals, constituting an independent new study equal in size to this one and so we don't think it is appropriate to call this a limitation of the current study. However, we do now make it clearer that adjacent pgACC/32 has been studied previously (see above).

2. Given the autonomic effects demonstrated, it is surprising not to see any metabolic changes in the dorsal ACC/Mid-cingulate, midbrain/brainstem (parabrachial nucleus, PAG). There are multiple human studies that would implicate those regions in the autonomic behaviors evoked here that might be reviewed and or discussed (see papers by Critchley, Riva Posse, Strick, among others).

As discussed above in reply to Reviewer 1 (point 7), the neuroimaging study does have low power owing to the number of subjects. Thus, it is possible that the autonomic effects are due to altered activity downstream of the amygdala/hypothalamus. We can only highlight, however, those areas that do show altered activity following sgACC/25. We do now point this out explicitly in the discussion (see above).

3. Given the focus of previous overactivation studies in the sgACC, do animals in the stress condition with sgACC overactivation demonstrate any of the anhedonic behaviors previously described? Are there methods to measure both within the same session? Humans can be both anxious and anhedonic; how might that be modelled using this probe?

This is an interesting question but one we didn't explicitly address here. We agree that it would seem likely given that the manipulation can induce both effects when given independently, and both anxious and anhedonic phenotypes can co-occur in humans with stress related disorders. Our PET study here and previously² would suggest that different networks are engaged following overactivation in threatening versus appetitive contexts, respectively, so the possibility exists that activation of sgACC/25 appetitive networks could occur independently of sgACC/25 threat-related networks. Indeed, this is something we'd like to pursue in the future.

We have previously used an approach-avoidance touchscreen task to measure conflict between reward and punishment. In this task, sgACC/25 over-activation enhanced punishment avoidance in the presence of reward¹⁶. However, whether this was driven specifically by increased sensitivity to punishment or reduced sensitivity to reward, or both was not assessed but could be in the future.

We have elaborated on this point at the end of the discussion. See our replies to point 7.

4. Lateralized PET findings. Were injections in the SCC unilateral or bilateral? The scans show clear laterality that is difficult to interpret (i.e., Left cortex/Right amygdala). Some discussion of laterality especially given the vagal findings seems warranted.

The injections into sgACC/25 were bilateral. We have mentioned this in the methods section but also added this clarification at the start of the results:

An overview of the marmosets used in these experiments is shown in **Table 1** with an experimental timeline in **Fig. 1A** and illustrations of three of the main paradigms in **Fig. 1B-D**. Histological assessment of cannula placement is shown in **Fig. 1E** together with estimated spread of infusions. **All infusions into sgACC/25 were bilateral.**

The laterality is indeed interesting. The cortex effects are right sided whereas the amygdala effects are on the left side as the images are reversed in conventional PET displays. Lateralization of effects have been demonstrated in many studies previously (especially of the amygdala), and it is difficult to address the causal implications of this, short of cannulating these downstream regions and manipulating only one side to compare the effects. We felt a detailed discussion of lateralization was beyond the scope of the manuscript especially given the small numbers of subjects in this PET study.

5. PET images are very small and hard to see; images might be larger so one can appreciate the anatomical location of the metabolic changes.

We have increased the size of the images. Note that the images are in high resolution so it should be possible to zoom into the image and see the areas of activation/de-activation more easily.

6. Might include discussion of other potential stress models that might address enhanced negative affect in depression more directly (in contrast to fear, threat, anxiety). Would be curious how a learned helplessness stress model or mild chronic stress would be impacted by sgACC overactivation.

We should make clear that in this study we were measuring the effects of sgACC/25 over-activity on acute reactivity to threat specifically and we were not trying to model a state of chronic stress per se. Moreover, it would be difficult to combine acute activation of sgACC/25 with such chronic models as proposed. Indeed, chronic stress may well induce chronic over-activation of sgACC/25. Nevertheless, the reviewer does raise the issue of comparability between acute manipulations performed here to the chronic nature of stress and how it can lead ultimately to stress-related disorders as expanded upon by the reviewer in their point 7. See our answer to address these issues below in point 7.

7. As both depression and anxiety are chronic conditions; the discussion might expand on the utility of this model for inducing sustained state changes appropriate for understanding pervasive anhedonia and or enhanced negative affect seen in human disorders.

The use of chronic mild stress is beyond the scope of this marmoset study. Nevertheless, we do agree that it is appropriate to highlight the acute nature of the effects we have induced here and the more chronic effects in stress-related disorders. We have expanded the discussion to include reference to these points as future directions at the end:

The work presented here lays the fundamental groundwork for several exciting future avenues exploring prefrontal contributions to stress-related disorders. First, the manipulations described in this study were acute whereas depression and anxiety are chronic conditions, so the effects of sustained sgACC/25 over-activity should be ascertained by using, for example, viral-mediated technologies. These techniques could also dissect out the selective sgACC/25 neural networks that may underlie the blunted appetitive and heightened aversive responses induced by over-activation which can co-exist in people with depression. Importantly, we can also identify the nature of the hierarchical interactions between higher-order prefrontal areas and sgACC/25 in the overall control of reward and threat-related behaviors.

References

1. Wallis, C. U., Cardinal, R. N., Alexander, L., Roberts, A. C. & Clarke, H. F. Opposing roles of primate areas 25 and 32 and their putative rodent homologs in the regulation of negative emotion. *Proc. Natl. Acad. Sci. U.S.A.* **114**, E4075–E4084 (2017).
2. Alexander, L. *et al.* Fractionating Blunted Reward Processing Characteristic of Anhedonia by Over-Activating Primate Subgenual Anterior Cingulate Cortex. *Neuron* **101**, 307–320.e6 (2019).
3. Mayberg, H. S. Limbic-cortical dysregulation: a proposed model of depression. *J Neuropsychiatry Clin Neurosci* **9**, 471–481 (1997).
4. Lane, R. D. *et al.* Neural correlates of heart rate variability during emotion. *Neuroimage* **44**, 213–222 (2009).
5. Allen, B., Jennings, J. R., Gianaros, P. J., Thayer, J. F. & Manuck, S. B. Resting high-frequency heart rate variability is related to resting brain perfusion. *Psychophysiology* **52**, 277–287 (2015).
6. Lane, R. D. *et al.* Subgenual anterior cingulate cortex activity covariation with cardiac vagal control is altered in depression. *Journal of Affective Disorders* **150**, 565–570 (2013).
7. Gianaros, P. J., Onyewuenyi, I. C., Sheu, L. K., Christie, I. C. & Critchley, H. D. Brain systems for baroreflex suppression during stress in humans. *Hum Brain Mapp* **33**, 1700–1716 (2011).
8. Barbas, H., Saha, S., Rempel-Clower, N. & Ghashghaei, T. Serial pathways from primate prefrontal cortex to autonomic areas may influence emotional expression. *BMC Neurosci* **4**, 25 (2003).
9. Joyce, M. K. P. & Barbas, H. Cortical Connections Position Primate Area 25 as a Keystone for Interoception, Emotion, and Memory. *J. Neurosci.* **38**, 1677–1698 (2018).
10. Chiba, T., Kayahara, T. & Nakano, K. Efferent projections of infralimbic and prelimbic areas of the medial prefrontal cortex in the Japanese monkey, *Macaca fuscata*. *Brain Res.* **888**, 83–101 (2001).
11. Reekie, Y. L., Braesicke, K., Man, M. S. & Roberts, A. C. Uncoupling of behavioral and autonomic responses after lesions of the primate orbitofrontal cortex. *Proc. Natl. Acad. Sci. U.S.A.* **105**, 9787–9792 (2008).
12. Ramirez-Mahaluf, J. P., Perramon, J., Otal, B., Villoslada, P. & Compte, A. Subgenual anterior cingulate cortex controls sadness-induced modulations of cognitive and emotional network hubs. *Scientific Reports* **8**, 8566 (2018).
13. Bob, P., Susta, M., Gregusova, A. & Jasova, D. Dissociation, cognitive conflict and nonlinear patterns of heart rate dynamics in patients with unipolar depression. *Progress in Neuro-Psychopharmacology and Biological Psychiatry* **33**, 141–145 (2009).
14. Liberzon, I. *et al.* Altered Central μ -Opioid Receptor Binding After Psychological Trauma. *Biological Psychiatry* **61**, 1030–1038 (2007).
15. Pardo, J. V. *et al.* Chronic vagus nerve stimulation for treatment-resistant depression decreases resting ventromedial prefrontal glucose metabolism. *Neuroimage* **42**, 879–889 (2008).
16. Wallis, C. U., Cockcroft, G. J., Cardinal, R. N., Roberts, A. C. & Clarke, H. F. Hippocampal Interaction With Area 25, but not Area 32, Regulates Marmoset Approach–Avoidance Behavior. *Cereb Cortex* **29**, 4818–4830 (2019).

Reviewers' Comments:

Reviewer #1:

Remarks to the Author:

I'm content with the authors' responses to my questions and concerns.

Asif Ghazanfar

Reviewer #2:

Remarks to the Author:

The authors present interesting and compelling findings. The paper has been enhanced by the changes made in the text in response to earlier comments, particularly around mechanisms. One minor point, which I return to is selectively calling a drug with multiple actions 'antidepressant' without stronger arguments as to why the effects observed at the doses given related to treatment of depression. The patented molecule Esketamine has indeed been licenced by the FDA (USA) and MHRA (UK) for antidepressant use as a nasal spray. Racemic ketamine (Ketavet), used here, has not. Clearly though, human studies of Ketamine have highlighted anxiolytic and antidepressant effects, though ketamine is however in the BNF for human use as an anaesthetic agent. As noted the intramuscular dose used in this study of nonhuman primate is lower than one would use for anaesthetic induction and effective sedation in humans and is in a range associated with memory disturbance, subjective affective changes and, in depressed people, antidepressant effects. I think the use of the term 'the antidepressant ketamine' is too selective, assumes (and ignores) a lot, and underplays the drug's other actions.

Reviewer #3:

Remarks to the Author:

The authors have provided a thoughtful response to all of the comments and questions raised in the original review. The changes improve the readability and clarity of the manuscript and its findings. I have no further questions, suggestions or concerns.

Helen Mayberg